# Exploiting Numerical Sparsity for Efficient Learning : Faster Eigenvector Computation and Regression

**Neha Gupta**
Department of Computer Science
Stanford University
Stanford, CA USA
nehagupta@cs.stanford.edu

**Aaron Sidford**
Department of Management Science and Engineering
Stanford University
Stanford, CA USA
sidford@stanford.edu

## Abstract

In this paper, we obtain improved running times for regression and top eigenvector computation for numerically sparse matrices. Given a data matrix $\boldsymbol{A} \in \mathbb{R}^{n \times d}$ where every row $a \in \mathbb{R}^d$ has $\|a\|_2^2 \leq L$ and numerical sparsity at most $s$, i.e. $\|a\|_1^2 / \|a\|_2^2 \leq s$, we provide faster algorithms for these problems in many parameter settings.

For top eigenvector computation, we obtain a running time of $\tilde{O}(nd + r(s + \sqrt{rs})/\text{gap}^2)$ where $\text{gap} > 0$ is the relative gap between the top two eigenvectors of $\boldsymbol{A}^\top \boldsymbol{A}$ and $r$ is the stable rank of $\boldsymbol{A}$. This running time improves upon the previous best unaccelerated running time of $O(nd + rd/\text{gap}^2)$ as $r \leq d$ and $s \leq d$.

For regression, we obtain a running time of $\tilde{O}(nd + (nL/\mu)\sqrt{snL/\mu})$ where $\mu > 0$ is the smallest eigenvalue of $\boldsymbol{A}^\top \boldsymbol{A}$. This running time improves upon the previous best unaccelerated running time of $\tilde{O}(nd + nLd/\mu)$. This result expands the regimes where regression can be solved in nearly linear time from when $L/\mu = \tilde{O}(1)$ to when $L/\mu = \tilde{O}(d^{2/3}/(sn)^{1/3})$.

Furthermore, we obtain similar improvements even when row norms and numerical sparsities are non-uniform and we show how to achieve even faster running times by accelerating using approximate proximal point [9] / catalyst [15]. Our running times depend only on the size of the input and natural numerical measures of the matrix, i.e. eigenvalues and $\ell_p$ norms, making progress on a key open problem regarding optimal running times for efficient large-scale learning.

## 1   Introduction

Regression and top eigenvector computation are two of the most fundamental problems in learning, optimization, and numerical linear algebra. They are central tools for data analysis and of the simplest problems in a hierarchy of complex machine learning computational problems. Consequently, developing provably faster algorithms for these problems is often a first step towards deriving new theoretically motivated algorithms for large scale data analysis.

Both regression and top eigenvector computation are known to be efficiently reducible [10] to the more general and prevalent finite sum optimization problem of minimizing a convex function $f$ decomposed into the sum of $m$ functions $f_1, ..., f_m$, i.e. $\min_{x \in \mathbb{R}^n} f(x)$ where $f(x) = \frac{1}{m} \sum_{i \in [m]} f_i(x)$. This optimization problem encapsulates a variety of learning tasks where we have data points $\{(a_1, b_1), (a_2, b_2), \cdots, (a_n, b_n)\}$ corresponding to feature vectors $a_i$, labels $b_i$, and we wish to find the predictor $x$ that minimizes the average loss of predicting $b_i$ from $a_i$ using $x$, denoted by $f_i(x)$.

Given the centrality of this problem to machine learning and optimization, over the past few years there have been extensive research efforts to design new provably efficient methods for solving this problem [12, 9, 13, 6, 20]. Using a variety of sampling techniques, impressive running time improvements have been achieved. The emphasis in this line of work has been on improving the dependence on the number of gradient evaluations of the $f_i$ that need to be performed, i.e. improving dependence on $m$, as well as improving the dependence on other problem parameters.

Much less studied is the question of what structural assumptions on $f_i$ allow even faster running times to be achieved. A natural and fundamental question in this space, is when can we achieve faster running times by computing the gradients of $f_i$ approximately, thereby decreasing iteration costs. While there has been work on combining coordinate descent methods with these stochastic methods [13], in the simple cases of regression and top eigenvector computation these methods do not yield any improvement in iteration cost. More broadly we are unaware of previous work on linearly convergent algorithms with faster running times for finite sum problems through this approach.

In this paper, we advance our understanding of the computational power of subsampling gradients of the $f_i$ for the problems of top eigenvector computation and regression. In particular, we show that under assumptions of numerical sparsity of the input matrix we can achieve provably faster algorithms and new nearly linear time algorithms for a broad range of parameters. We achieve our result by applying coordinate sampling techniques to Stochastic Variance Reduced Gradient Descent (SVRG) [13, 12], a popular tool for finite sum optimization, along with linear algebraic data structures (in the case of eigenvector computation) that we believe may be of independent interest.

The results in this paper constitute an important step towards resolving a key gap in our understanding of optimal iterative methods for top eigenvector computation and regression. Ideally running times of these problems would depend only on the size of the input, e.g. the number of non-zero entries in the input data matrix, row norms, eigenvalues, etc. However, this is not the case for the current fastest regression algorithms as these methods work by picking rows of the matrix non-uniformly yielding expected iteration costs that depend on brittle weighted sparsity measures (which for simplicity are typically instead stated in terms of the maximum sparsity among all rows, see Section 1.4.1). This causes particularly unusual running times for related problems like nuclear norm estimation [17].

This paper takes an important step towards resolving this problem by providing running times for top eigenvector computation and regression that depend only on the size of the input and natural numerical quantities like eigenvalues, $\ell_1$-norms, $\ell_2$-norms, etc. While our running times do not strictly dominate those based on the sparsity structure of the input (and it is unclear if such running times are possible), they improve upon the previous work in many settings. Ultimately, we hope this paper provides useful tools for even faster algorithms for solving large scale learning problems.

## 1.1 The Problems

Throughout this paper we let $\boldsymbol{A} \in \mathbb{R}^{n \times d}$ denote a data matrix with rows $a_1, ..., a_n \in \mathbb{R}^d$. We let $\mathrm{sr}(\boldsymbol{A}) \stackrel{\text{def}}{=} \|\boldsymbol{A}\|_F^2 / \|\boldsymbol{A}\|_2^2$ denote the stable rank of $\boldsymbol{A}$ and we let $\mathrm{nnz}(\boldsymbol{A})$ denote the number of non-zero entries in $\boldsymbol{A}$. For symmetric $\boldsymbol{M} \in \mathbb{R}^{d \times d}$ we let $\lambda_1(\boldsymbol{M}) \geq \lambda_2(\boldsymbol{M}) \geq ... \geq \lambda_d(\boldsymbol{M})$ denote its eigenvalues, $\|x\|_{\boldsymbol{M}}^2 = x^\top \boldsymbol{M} x$ and we let $\mathrm{gap}(\boldsymbol{M}) \stackrel{\text{def}}{=} (\lambda_1(\boldsymbol{M}) - \lambda_2(\boldsymbol{M}))/\lambda_1(\boldsymbol{M})$ denote its (relative) eigenvalue gap. For convenience we let $\mathrm{gap} \stackrel{\text{def}}{=} \mathrm{gap}(\boldsymbol{A}^\top \boldsymbol{A})$, $\lambda_1 \stackrel{\text{def}}{=} \lambda_1(\boldsymbol{A}^\top \boldsymbol{A})$, $\mu \stackrel{\text{def}}{=} \lambda_{\min} \stackrel{\text{def}}{=} \lambda_d(\boldsymbol{A}^\top \boldsymbol{A})$, $\mathrm{sr} \stackrel{\text{def}}{=} \mathrm{sr}(\boldsymbol{A})$, $\mathrm{nnz}(\boldsymbol{A}) \stackrel{\text{def}}{=} \mathrm{nnz}$, $\kappa \stackrel{\text{def}}{=} \|\boldsymbol{A}\|_F^2 / \mu$ and $\kappa_{\max} \stackrel{\text{def}}{=} \lambda/\mu$. With this notation, we consider the following two optimization problems.

**Definition 1 (Top Eigenvector Problem)** *Find $v^* \in \mathbb{R}^d$ such that*

$$v^* = \operatorname*{argmax}_{x \in \mathbb{R}^d, \|x\|_2 = 1} x^\top \boldsymbol{A}^\top \boldsymbol{A} x$$

*We call $v$ an $\epsilon$-approximate solution to the problem if $\|v\|_2 = 1$ and $v^\top \boldsymbol{A}^\top \boldsymbol{A} v \geq (1 - \epsilon)\lambda_1(\boldsymbol{A}^\top \boldsymbol{A})$.*

**Definition 2 (Regression Problem)** *Given $b \in \mathbb{R}^n$ find $x^* \in \mathbb{R}^d$ such that*

$$x^* = \operatorname*{argmin}_{x \in \mathbb{R}^d} \|\boldsymbol{A}x - b\|_2^2$$

*Given initial $x_0 \in \mathbb{R}^d$, we call $x$ an $\epsilon$-approximate solution if $\|x - x^*\|_{\boldsymbol{A}^\top \boldsymbol{A}} \leq \epsilon \|x_0 - x^*\|_{\boldsymbol{A}^\top \boldsymbol{A}}$.*

Each of these are known to be reducible to the finite sum optimization problem. The regression problem is equivalent to the finite sum problem with $f_i(x) \stackrel{\text{def}}{=} (m/2)(a_i^\top x - b_i)^2$ and the top eigenvector problem is reducible with only polylogarithmic overhead to the finite sum problem with $f_i(x) \stackrel{\text{def}}{=} \lambda \|x - x_0\|_2^2 - (m/2)(a_i^\top(x - x_0))^2 + b_i^\top x$ for carefully chosen $\lambda$ and $x_0$ [10].

## 1.2 Our Results

In this paper, we provide improved iterative methods for top eigenvector computation and regression that depend only on regularity parameters and not the specific sparsity structure of the input. Rather than assuming uniform row sparsity as in previous work our running times depend on the numerical sparsity of rows of $A$, i.e. $s_i \stackrel{\text{def}}{=} \|a_i\|_1^2/\|a_i\|_2^2$, which is at most the row sparsity, but may be smaller.

Note that our results, as stated, are worse as compared to the previous running times which depend on the $\ell_0$ sparsity in some parameter regimes. For simplicity, we are stating our results in terms of only the numerical sparsity. However, when the number of zero entries in a row is small, we can always choose that row completely and not do sampling on it. This would lead to our results always as good as the previous results and stricly better in some parameter regimes.

### 1.2.1 Top Eigenvector Computation

For top eigenvector computation, we give an unaccelerated running time of $\tilde{O}(\text{nnz}(A) + 1/(\text{gap}^2)\sum_i(\|a_i\|_2^2/\lambda_1)(\sqrt{s_i} + \sqrt{\text{sr}(A)})\sqrt{s_i})$ and an accelerated running time of $\tilde{O}(\text{nnz}(A) + (\text{nnz}(A)^{\frac{3}{4}}/\sqrt{\text{gap}})(\sum_i(\|a_i\|_2^2/\lambda_1)(\sqrt{s_i} + \sqrt{\text{sr}(A)})\sqrt{s_i})^{\frac{1}{4}})$ as compared to the previous unaccelerated running time of $\tilde{O}(\text{nnz}(A) + \max_i \text{nnz}(a_i)\text{sr}(A)/\text{gap}^2)$ and accelerated iterative methods of $\tilde{O}(\text{nnz}(A)^{3/4}(\max_i \text{nnz}(a_i)(\text{sr}(A)/\text{gap}^2))^{1/4})$ respectively.

In the simpler case of uniform row norms $\|a_i\|_2^2 = \|a\|_2^2$ and uniform row sparsity $s_i = s$, our running time (unaccelerated) becomes $\tilde{O}(\text{nnz}(A) + (\text{sr}(A)/\text{gap}^2)(s + \sqrt{\text{sr}(A) \cdot s}))$. To understand the relative strength of our results, we give an example of one parameter regime where our running times are strictly better than the previous running times. When the rows are numerically sparse i.e. $s = O(1)$ although the number of $\text{nnz}(a_i) = d$, then our running time $\tilde{O}(\text{nnz}(A) + (\text{sr}(A)/\text{gap}^2)\sqrt{\text{sr}(A)})$ gives significant improvement over the previous best running time of $\tilde{O}(\text{nnz}(A) + d(\text{sr}(A)/\text{gap}^2))$ since $\text{sr}(A) \leq d$.

### 1.2.2 Regression

For regression we give an unaccelerated running time of $\tilde{O}(\text{nnz}(A) + \sqrt{\kappa}\sum_i \sqrt{s_i}\|a_i\|_2^2/\mu)$ and an accelerated running time of $\tilde{O}(\text{nnz}(A)^{2/3}\kappa^{1/6}(\sum_{i\in[n]} \sqrt{s_i}\|a_i\|_2^2/\mu)^{1/3})$. Our methods improve upon the previous best unaccelerated iterative methods of $\tilde{O}(\text{nnz}(A) + \kappa \max_i \text{nnz}(a_i))$ and accelerated iterative methods of $\tilde{O}(\text{nnz}(A) + d\max_i \text{nnz}(a_i) + \sum_i(\|a_i\|_2/\sqrt{\mu})\sqrt{\sigma_i}(A)\max_i \text{nnz}(a_i))$ where $\sigma_i = \|a_i\|_{(A^\top A)^{-1}}^2$.

In the simpler case of uniform row norms $\|a_i\|_2^2 = \|a\|_2^2$ and uniform row sparsity $s_i = s$, our (unaccelerated) running time becomes $\tilde{O}(\text{nnz}(A) + \kappa^{3/2}\sqrt{s_i})$.

To understand the relative strength of our results, we give an example of one parameter regime where our running times are strictly better than the previous running times. Consider the case where $\kappa = o(d^2)$ and the rows are numerically sparse i.e. $s = O(1)$ but $\max_i \text{nnz}(a_i) = d$, consider the particular case of $\kappa = d^{1.5}$, then our running time is $\tilde{O}(\text{nnz}(A) + d^{2.25})$ whereas the SVRG running time for regression will be $\tilde{O}(\text{nnz}(A) + d^{2.5})$ and our running time is better in this case.

## 1.3 Overview of Our Approach

We achieve these results by carefully modifying known techniques for finite sum optimization problem to our setting. The starting point for our algorithms is Stochastic Variance Reduced Gradient Descent (SVRG) [12] a popular method for finite sum optimization. This method takes steps in the

direction of negative gradient in expectation and its convergence depends on a measure of variance of the steps.

We apply SVRG to our problems where we carefully subsample the entries of the rows of the data matrix so that we can compute steps that are the negative gradient in expectation in time possibly sublinear in the size of the row. There is an inherent issue in such a procedure, in that this can change the shape of variance. Previous sampling methods for regression ensure that the variance can be directly related to the function error, whereas here such sampling methods give $\ell_2$ error, the bounding of which in terms of function error can be expensive.

It is unclear how to completely avoid this issue and we leave this as future work. Instead, to mitigate this issue we provide several techniques for subsampling that ensure we can obtain significant decrease in this $\ell_2$ error for small increases in the number of samples we take per row (See Section 3). Here we crucially use that we have bounds on the numerical sparsity of rows of the data matrix and prove that we can use this to quantify this decrease.

Formally, the sampling problem we have for each row is as follows. For each row $a_i$ at any point we may receive some vector $x$ and need to compute a random vector $g$ with $\mathbb{E}[g] = a_i a_i^\top x$ and with $\mathbb{E}\|g\|_2^2$ sufficiently bounded. Ideally, we would have that $\mathbb{E}\|g\|_2^2 \leq \alpha(a_i^\top x)$ for some value of $\alpha$, as previous methods do. However, instead we settle for a bound of the form $\mathbb{E}\|g\|_2^2 \leq \alpha(a_i^\top x) + \beta\|x\|_2^2$. Our sampling schemes for this problem works as follows: For the outer $a_i$, we sample from the coordinates with probability proportional to the coordinate's absolute value, we take a few (more than 1) samples to control the variance (Lemma 4). For the approximation of $a_i^\top x$, we always take the dot product of $x$ with large coordinates of $a_i$ and we sample from the rest with probability proportional to the squared value of the coordinate of $a_i$ and take more than one sample to control the variance (Lemma 5).

Carefully controlling the number of samples we take per row and picking the right distribution over rows gives our bounds for regression. For eigenvector computation the same broad techniques work but to keep the iteration costs down but a little more care needs to be taken due to the structure of $f_i(x) \stackrel{\text{def}}{=} \lambda\|x - x_0\|_2^2 - (m/2)(a_i^\top(x - x_0))^2 + b_i^\top x$. Interestingly, for eigenvector computation the penalty from $\ell_2$ error is in some sense smaller due to the structure of the objective.

## 1.4 Previous Results

Here we briefly cover previous work on regression and eigenvector computation (Section 1.4.1), sparse finite sum optimization (Section 1.4.2), and matrix entrywise sparsification (Section 1.4.3).

### 1.4.1 Regression and Eigenvector Algorithms

There is an extensive amount of work on regression, eigenvector computation, and finite sum optimization with far too many results to state but we have tried to include the algorithms with the best known running times. The results for top eigenvector computation are stated in Table 1 and the results for regression are stated in Table 2. The algorithms work according to the weighted $\ell_0$ sparsity measure of all rows and do not take into account the numerical sparsity which is a natural parameter to state the running times in and is strictly better than the $\ell_0$ sparsity.

### 1.4.2 Sparsity Structure

There has been some prior work on attempting to improve for sparsity structure. Particularly relevant is the work of [13] on combining coordinate descent and sampling schemes. This paper picks unbiased estimates of the gradient at each step by first picking a function and then picking a random coordinate whose variance decreases as time increases. Unfortunately, for regression and eigenvector computation computing a partial derivative is as expensive as computing the gradient and hence, this method does not give improved running times for regression and top eigenvector computation.

### 1.4.3 Entrywise Sparsification

Another natural approach to yielding the results of this paper would be to simply subsample the entries of $A$ beforehand and use this as a preconditioner to solve the problem. There have been multiple works on such entrywise sparsification and in Table 3 we provide them. If we optimistically

Table 1: Previous results for computing $\epsilon$-approximate top eigenvector (Definition 1).

| Algorithm | Runtime | Runtime with uniform row norms and sparsity |
|---|---|---|
| Power Method | $\tilde{O}\left(\frac{\text{nnz}}{\text{gap}}\right)$ | $\tilde{O}\left(\frac{nd}{\text{gap}}\right)$ |
| Lanczos Method | $\tilde{O}\left(\frac{\text{nnz}}{\sqrt{\text{gap}}}\right)$ | $\tilde{O}\left(\frac{nd}{\sqrt{\text{gap}}}\right)$ |
| Fast subspace embeddings + Lanczos method [7] | $\tilde{O}\left(\text{nnz} + \frac{d\cdot\text{sr}}{\max\{\text{gap}^{2.5},\epsilon,\epsilon^{2.5}\}}\right)$ | $\tilde{O}\left(nd + \frac{d\cdot\text{sr}}{\max\{\text{gap}^{2.5},\epsilon,\epsilon^{2.5}\}}\right)$ |
| SVRG (assuming bounded row norms and warm start) [21] | $\tilde{O}\left(\text{nnz} + \frac{d\cdot\text{sr}^2}{\text{gap}^2}\right)$ | $\tilde{O}\left(nd + \frac{d\cdot\text{sr}^2}{\text{gap}^2}\right)$ |
| Shift & Invert Power method with SVRG [10] | $\tilde{O}\left(\text{nnz} + \frac{d\cdot\text{sr}}{\text{gap}^2}\right)$ | $\tilde{O}\left(nd + \frac{d\cdot\text{sr}}{\text{gap}^2}\right)$ |
| Shift & Invert Power method with Accelerated SVRG [10] | $\tilde{O}\left(\text{nnz} + \frac{\text{nnz}^{3/4}(d\cdot\text{sr})^{1/4}}{\sqrt{\text{gap}}}\right)$ | $\tilde{O}\left(nd + \frac{(nd)^{3/4}(d\cdot\text{sr})^{1/4}}{\sqrt{\text{gap}}}\right)$ |
| This paper | $\tilde{O}(\text{nnz} + \frac{1}{\text{gap}^2\lambda_1}\sum_i \|a_i\|_2^2 (\sqrt{s_i}+\sqrt{\text{sr}})\sqrt{s_i})$ | $\tilde{O}(nd + \frac{\text{sr}}{\text{gap}^2}(\sqrt{s}+\sqrt{\text{sr}})\sqrt{s})$ |
| This paper | $\tilde{O}(\text{nnz} + \frac{\text{nnz}^{\frac{3}{4}}}{\sqrt{\text{gap}}}(\sum_i \frac{\|a_i\|_2^2}{\lambda_1}(\sqrt{s_i}+\sqrt{\text{sr}})\sqrt{s_i}))^{\frac{1}{4}})$ | $\tilde{O}(nd + \frac{(nd)^{\frac{3}{4}}}{\sqrt{\text{gap}}}\text{sr}^{\frac{1}{4}}(s+\sqrt{\text{sr}\cdot s})^{\frac{1}{4}})$ |

Table 2: Previous results for solving approximate regression (Definition 2).

| Algorithm | Runtime | Runtime with uniform row norms and sparsity |
|---|---|---|
| Gradient Descent | $\tilde{O}(\text{nnz}\cdot\kappa_{\max})$ | $\tilde{O}(nd\kappa_{\max})$ |
| Conjugate Gradient Descent | $\tilde{O}(\text{nnz}\sqrt{\kappa_{\max}})$ | $\tilde{O}(nd\sqrt{\kappa_{\max}})$ |
| SVRG [12] | $\tilde{O}(\text{nnz} + \kappa d)$ | $\tilde{O}(nd + \kappa d)$ |
| Accelerated SVRG [4, 9, 15] | $\tilde{O}(\text{nnz} + \sqrt{n\kappa}d)$ | $\tilde{O}(nd + \sqrt{n\kappa}d)$ |
| Accelerated SVRG with leverage score sampling [3] | $\tilde{O}(\text{nnz} + d\max_i \text{nnz}(a_i) + \sum_i \frac{\|a_i\|_2}{\sqrt{\mu}}\cdot \sqrt{\sigma_i(\boldsymbol{A})}\max_i \text{nnz}(a_i))$ | $\tilde{O}(nd + d^2 + \sqrt{\kappa}\cdot d^{3/2})$ |
| This paper | $\tilde{O}(\text{nnz} + \sqrt{\kappa}\sum_i \frac{\|a_i\|_2^2}{\mu}\sqrt{s_i})$ | $\tilde{O}(nd + \sqrt{\kappa^3}\sqrt{s})$ |
| This paper | $\tilde{O}(\text{nnz}^{2/3}\kappa^{1/6}(\sum_{i\in[n]} \frac{\|a_i\|_2^2}{\mu}\sqrt{s_i})^{1/3})$ | $\tilde{O}((nd)^{2/3}\kappa^{1/2}s^{1/6})$ |

compare them to our approach, by supposing that their sparsity bounds are uniform (i.e. every row has the same sparsity) and bound its quality as a preconditioner the best of these would give bounds of $\tilde{O}(\text{nnz}(\boldsymbol{A}) + \lambda_{\max}\|\boldsymbol{A}\|_F^4/\lambda_{\min}^3)$ [14] and $\tilde{O}(\text{nnz}(\boldsymbol{A}) + \sqrt{\lambda_{\max}}\|\boldsymbol{A}\|_F^2\sum_i \sqrt{s_i}\|a_i\|_2/\sqrt{n}\lambda_{\min}^2)$ [5] and $\tilde{O}(\text{nnz}(\boldsymbol{A}) + \|\boldsymbol{A}\|_F^2\sum_i s_i\|a_i\|_2^2\lambda_{\max}/n\lambda_{\min}^3)$ [1] for regression. Bound obtained by [14] depends on the the condition number square and does not depend on the numerical sparsity structure of the matrix. Bound obtained by [5] is worse as compared to our bound when compared with matrices having equal row norms and uniform sparsity. Our running time for regression is $\tilde{O}(\text{nnz}(\boldsymbol{A}) + \sqrt{\kappa}\sum_i(\|a_i\|_2^2/\mu)\sqrt{s_i})$. Our results are not always comparable to that by [1]. Assuming uniform sparsity and row norms, we get that Our runtime/Runtime by $[1] = (\lambda_{\min}n)/(\sqrt{s}\lambda_{\max}\sqrt{\kappa})$. Depending on the values of the particular parameters, the ratio can be both greater or less than 1 and hence, the results are incomparable. Our results are always better than that obtained by [14].

## 2  Notation

**Vector Properties:** For $a \in \mathbb{R}^d$, let $s(a) = \|a\|_1^2/\|a\|_2^2$ denote the numerical sparsity. For $c \in \{1, 2, \ldots, d\}$, let $(\Pi_c(a))_i = a_i$ if $i \in S$ where $S$ is a set of the $c$ largest coordinates of $a$ in absolute value and 0 otherwise and $\bar{\Pi}_c(a) = a - \Pi_c(a)$. Let $I_c(a)$ denote the set of indices with the $c$ largest coordinates of $a$ in absolute value and $\bar{I}_c(a) = [d] \setminus I_c(a)$ i.e. everything except the top $c$ co-ordinates. Let $\hat{e}_j$ denote the ith basis vector i.e. $(\hat{e}_j)_i = 1$ if $i = j$ and 0 otherwise.

**Other:** Let $[d]$ denote the set $\{1, 2, \ldots, d\}$. We use $\tilde{O}$ notation to hide polylogarithmic factors in the input parameters and error rates. Refer to Section 1.1 for other definitions.

## 3  Sampling techniques

In this section we provide our key tools for sampling from a matrix for both regression and eigenvector computation. First, we provide a technical lemma on numerical sparsity that we will use throughout our analysis. Then, we provide and analyze the sampling distribution we use to sample from our matrix for SVRG. We use the same distribution for both the applications, regression and eigenvector computation and provide some of the analysis of properties of this distribution. All proofs in this section are differed to Appendix B.1.

We begin with a lemma at the core of the proofs of our sampling techniques. The lemma essentially states that for a numerically sparse vector, most of the $\ell_2$-mass of the vector is concentrated in its top few coordinates. Consequently, if a vector is numerically sparse then we can remove a few big coordinates from it and reduce its $\ell_2$ norm considerably. Later, in our sampling schemes, we will use this lemma to bound the variance of sampling a vector.

**Lemma 3 (Numerical Sparsity)**  *For $a \in \mathbb{R}^d$ and $c \in [d]$, we have $\|\bar{\Pi}_c(a)\|_2^2 \leq s(a)\|a\|_2^2/c$.*

The following lemmas state the sampling distribution that we use for sampling the gradient function in SVRG. Basically, since we want to approximate the gradient of $f(x) = \frac{1}{2}x A^\top A x - b^\top x$ i.e. $A^\top A x - b$, we would like to sample $A^\top A x = \sum_{i \in [n]} a_i a_i^\top x$.

We show how to perform this sampling and analyze it in several steps. In Lemma 4 we show how to sample from $a$ and then in Lemma 5 we show how to sample from $a^\top x$. In Lemma 6 we put these together to sample from $aa^\top x$ and in Lemma 7 we put it all together to sample from $A^\top A$. The distributions and our guarantees on them are stated below.

| **Algorithm 1:** Samplevec$(a, c)$ | **Algorithm 2:** Sampledotproduct$(a, x, c)$ |
|---|---|
| 1: **for** $t = 1 \ldots c$ (i.i.d. trials) **do** | 1: **for** $t = 1 \ldots c$ (i.i.d. trials) **do** |
| 2:   randomly sample indices $j_t$ with | 2:   randomly sample indices $j_t$ with |
| 3:   $\Pr(j_t = j) = p_j = \frac{|a_j|}{\|a\|_1} \quad \forall j \in [d]$ | 3:   $\Pr(j_t = j) = p_j = \frac{a_j^2}{\|\bar{\Pi}_c(a)\|_2^2} \quad \forall j \in \bar{I}_c(a)$ |
| 4: **end for** | 4: **end for** |
| 5: Output $\frac{1}{c}\sum_{t=1}^{c} \frac{a_{j_t}}{p_{j_t}} \hat{e}_{j_t}$ | 5: Output $\Pi_c(a)^\top x + \frac{1}{c}\sum_{t=1}^{c} \frac{a_{j_t} x_{j_t}}{p_{j_t}}$ |

| **Algorithm 3:** Samplerankonemat$(a, x, c)$ | **Algorithm 4:** Samplemat$(A, x, k)$ |
|---|---|
| 1: $(\widehat{a})_c = $ Samplevec$(a, c)$ | 1: $c_i = \sqrt{s_i} \cdot k \quad \forall i \in [n]$ |
| 2: $\widehat{(a^\top x)}_c = $ Sampledotproduct$(a, x, c)$ | 2: $M = \sum_i \|a_i\|_2^2 (1 + \frac{s_i}{c_i})$ |
| 3: Output $(\widehat{a})_c \widehat{(a^\top x)}_c$ | 3: Select a row index $i$ with probability |
| | $\quad p_i = \frac{\|a_i\|_2^2}{M}(1 + \frac{s_i}{c_i})$ |
| | 4: $\widehat{(a_i a_i^\top x)}_{c_i} = $ Samplerankonemat$(a_i, x, c_i)$ |
| | 5: Output $\frac{1}{p_i}\widehat{(a_i a_i^\top x)}_{c_i}$ |

**Lemma 4 (Stochastic Approximation of** $a$**)**  *Let $a \in \mathbb{R}^d$ and $c \in \mathbb{N}$ and let our estimator $(\hat{a})_c = $ Samplevec$(a, x)$ (Algorithm 1) Then,*

$$\mathbb{E}[(\hat{a})_c] = a \ \text{ and } \ \mathbb{E}\left[\|(\hat{a})_c\|_2^2\right] \leq \|a\|_2^2 \left(1 + \frac{s(a)}{c}\right)$$

**Lemma 5 (Stochastic Approximation of** $a^\top x$**)**  *Let $a, x \in \mathbb{R}^d$ and $c \in [d]$, and let our estimator be defined as $\widehat{(a^\top x)}_c = $ Sampledotproduct$(a, x, c)$ (Algorithm 2) Then,*

$$\mathbb{E}[\widehat{(a^\top x)}_c] = a^\top x \ \text{ and } \ \mathbb{E}\left[\widehat{(a^\top x)}_c^2\right] \leq (a^\top x)^2 + \frac{1}{c}\|\bar{\Pi}_c(a)\|_2^2 \|x\|_2^2$$

**Lemma 6 (Stochastic Approximation of $aa^\top x$)** *Let $a, x \in \mathbb{R}^d$ and $c \in [d]$, and the estimator be defined as $\widehat{(aa^\top x)}_c = \texttt{Samplerankonemat}(a, x, c)$ (Algorithm 3) Then,*

$$\mathbb{E}[\widehat{(aa^\top x)}_c] = aa^\top x \ \text{ and } \ \mathbb{E}\left[\|\widehat{(aa^\top x)}_c\|_2^2\right] \leq \|a\|_2^2 \left(1 + \frac{s(a)}{c}\right) \left((a^\top x)^2 + \frac{s(a)}{c^2}\|a\|_2^2\|x\|_2^2\right)$$

**Lemma 7 (Stochastic Approximation of $A^\top A x$ )** *Let $A \in \mathbb{R}^{n \times d}$ with rows $a_1, a_2, \ldots, a_n$ and $x \in \mathbb{R}^d$ and let $\widehat{(A^\top Ax)}_k = \texttt{Samplemat}(A, x, k)$ (Algorithm 4) where $k$ is some parameter. Then,*

$$\mathbb{E}\left[\widehat{(A^\top Ax)}_k\right] = A^\top A x \ \text{ and } \ \mathbb{E}\left[\left\|\widehat{(A^\top Ax)}_k\right\|_2^2\right] \leq M\left(\|Ax\|_2^2 + \frac{1}{k^2}\|A\|_F^2\|x\|_2^2\right).$$

# 4  Applications

Using the framework of SVRG defined in Theorem 14 and the sampling techniques presented in Section 3, we now state how do we solve our problems of regression and top eigenvector computation.

## 4.1  Eigenvector computation

The classic method to estimate the top eigenvector of a matrix is to apply power method which involves starting with an initial vector $x_0$ and repeatedly multiplying the vector by $A^\top A$ which eventually leads to convergence of the vector to the top eigenvector of the matrix $A^\top A$ if top eigenvalue of the matrix is well separated from the other eigenvalues i.e. gap is large enough. The number of iterations required for convergence is $O(\log(\frac{d}{\epsilon})/\text{gap})$. However, this method can be very slow when the gap is small. If the gap is small, one thing that can be done to improve convergence rate is running the power method on a matrix $B^{-1}$ where $B = \lambda I - A^\top A$. $B^{-1}$ has the same largest eigenvector as $A^\top A$ and the eigenvalue gap is $(\frac{1}{\lambda - \lambda_1} - \frac{1}{\lambda - \lambda_2})/\frac{1}{\lambda - \lambda_1} = \frac{1}{2}$ if $\lambda \approx (1 + \text{gap})\lambda_1$ and thus we get a constant eigenvalue gap. Hence, if we have a rough estimate of the largest eigenvalue of the matrix, we can get the gap parameter as roughly constant. Section 6 of [10] shows how we can get such an estimate based on the gap free eigenvalue estimation algorithm by [16] in running time dependent on the linear system solver of $B$ ignoring some additional polylogarithmic factors. However, doing power iteration on $B^{-1}$ requires solving linear systems on $B$ whose condition number now depends on $1/\text{gap}$ and thus, solving linear system on $B$ would become expensive. [10] showed how we can solve the linear systems in $B$ faster by using SVRG [12] and achieved a better overall running time for top eigenvector computation. The formal theorem statement is differed to Theorem 17 in the appendix.

We simply use this framework for solving the eigenvector problem using SVRG and on the top of that, give different sampling scheme for SVRG for $B^{-1}$ which reduces the runtime for numerically sparse matrices. Basically, we use the sampling scheme presented in Lemma 7. The following lemma states the variance bound that we get for the gradient updates for SVRG for the top eigenvector computation problem. This will be used to obtain a bound on the solving of linear systems in $B = \lambda I - A^\top A$ which will be ultimately used in solving the approximate topeigen vector problem.

**Lemma 8 (Variance bound for eigenvector computation)** *Let $\nabla g(x) = \lambda x - \widehat{(A^\top Ax)}_k$ where $\widehat{(A^\top Ax)}_k$ is the estimator of $A^\top A x$ defined in Lemma 7, and $k = \sqrt{\text{sr}(A)}$, then we get*

$$\mathbb{E}[\nabla g(x)] = (\lambda I - A^\top A)x \ \text{ and } \ \mathbb{E}\left[\|\nabla g(x) - \nabla g(x^*)\|_2^2\right] \leq (f(x) - f(x^*))8M/\text{gap}$$

*with average time taken in calculating $\nabla g(x), T = \sum_i \|a_i\|_2^2 \left(s_i + \sqrt{s_i \text{sr}(A)}\right)/M$ where $M = \sum_i \|a_i\|_2^2 \left(1 + \sqrt{\frac{s_i}{\text{sr}(A)}}\right)$ and $f(x) = \frac{1}{2}x^\top B x - b^\top x$*

Now, using the variance of the gradient estimators and per iteration running time $T$ obtained in Lemma 8 along with the framework of SVRG [12] (defined in Theorem 14), we can get constant multiplicative decrease in the error in solving linear systems in $B = \lambda I - A^\top A$ in total running time

$O(\mathrm{nnz}(\boldsymbol{A}) + \frac{2}{\mathrm{gap}^2\lambda_1(\boldsymbol{A}^\top\boldsymbol{A})} \sum_i \|a_i\|_2^2 (\sqrt{s_i} + \sqrt{\mathrm{sr}(\boldsymbol{A})})\sqrt{s_i})$ assuming we have a crude approximation to the top eigenvector and eigenvalue which we have already discussed we can get. The formal theorem statement (Theorem 18) and proof are differed to the appendix. Now, using the linear system solver descibed above along with the shift and invert algorithmic framework, we get the following running time for top eigenvector computation problem. The proof appears in Appendix B.2.

**Theorem 9 (Numerically Sparse Top Eigenvector Computation Runtime)** *Linear system solver from Theorem 18 combined with the shift and invert framework from [10] stated in Theorem 17 gives an algorithm which computes $\epsilon$-approximate top eigenvector (Definition 1) in total running time*

$$O\left(\left(\mathrm{nnz}(\boldsymbol{A}) + \frac{1}{\mathrm{gap}^2\lambda_1} \sum_i \|a_i\|_2^2 \left(\sqrt{s_i} + \sqrt{\mathrm{sr}(\boldsymbol{A})}\right)\sqrt{s_i}\right) \cdot \left(\log^2\left(\frac{d}{\mathrm{gap}}\right) + \log\left(\frac{1}{\epsilon}\right)\right)\right)$$

Similarly, using the acceleration framework of [9] mentioned in Theorem 15 in the appendix along with the linear system solver runtime, we get the following accelerated running time for top eigenvector computation and the proof appears in Appendix B.2.

**Theorem 10 (Numerically Sparse Accelerated Top Eigenvector Computation Runtime)**
*Linear system solver from Theorem 18 combined with acceleration framework from [9] mentioned in Theorem 15 and shift and invert framework from [10] stated in Theorem 17 gives an algorithm which computes $\epsilon$-approximate top eigenvector (Definition 1) in total running time*

$$\tilde{O}\left(\mathrm{nnz}(\boldsymbol{A}) + \frac{\mathrm{nnz}(\boldsymbol{A})^{3/4}}{\sqrt{\mathrm{gap}}} \left(\sum_i \frac{\|a_i\|_2^2}{\lambda_1}\left(\sqrt{s_i} + \sqrt{\mathrm{sr}(\boldsymbol{A})}\right)\sqrt{s_i}\right)^{1/4}\right) \text{ where } \tilde{O} \text{ hides a factor of}$$

$\left(\log^2\left(\frac{d}{\mathrm{gap}}\right) + \log\left(\frac{1}{\epsilon}\right)\right)\log\left(\frac{d}{\mathrm{gap}}\right).$

## 4.2 Linear Regression

In linear regression, we want to minimize $\frac{1}{2}\|\boldsymbol{A}x - b\|_2^2$ which is equivalent to minimizing $\frac{1}{2}x^\top\boldsymbol{A}^\top\boldsymbol{A}x - x^\top\boldsymbol{A}^\top b = \frac{1}{2}\sum_i x^\top a_i a_i^\top x - x^\top\boldsymbol{A}^\top b$ and hence, we can apply the framework of SVRG [12] (stated in Theorem 14) for solving it. However, instead of selecting the complete row for calculating the gradient, we only select a few entries from the row to achieve lower cost per iteration. In particular, we use the distribution defined in Lemma 7. Note that the sampling probabilities depend on $\lambda_d$ and we need to know a constant factor approximation of $\lambda_d$ for the scheme to work. For most of the ridge regression problems, we know a lower bound on the value of $\lambda_d$ and we can get an approximation by doing a binary search over all the values and paying an extra logarithmic factor. The following lemma states the sampling distribution which we use for approximating the true gradient and the corresponding variance that we obtain. The proof of this appears in Appendix B.2.

**Lemma 11 (Variance Bound for Regression)** *Let $\nabla g(x) = \widehat{(\boldsymbol{A}^\top\boldsymbol{A}x)}_k$ where $\widehat{(\boldsymbol{A}^\top\boldsymbol{A}x)}_k$ is the estimator for $\boldsymbol{A}^\top\boldsymbol{A}x$ defined in Lemma 7 and $k = \sqrt{\kappa}$, assuming $\kappa \leq d^2$ we get*

$$\mathbb{E}[\nabla g(x)] = \boldsymbol{A}^\top\boldsymbol{A}x \quad and \quad \mathbb{E}\left[\|\nabla g(x) - \nabla g(x^*)\|_2^2\right] \leq M(f(x) - f(x^*))$$

*with average time taken in calculating $\nabla g(x)$, $T = \frac{\sqrt{\kappa}}{M}\sum_{i\in[n]}\|a_i\|_2^2\sqrt{s_i}$ where $M = \sum_i \|a_i\|_2^2\left(1 + \sqrt{\frac{s_i}{\kappa}}\right)$ where $f(x) = \frac{1}{2}\|\boldsymbol{A}x - b\|_2^2$*

Using the variance bound obtained in Lemma 11 and the framework of SVRG stated in Theorem 14 for solving approximate linear systems, we show how we can obtain an algorithm for solving approximate regression in time which is faster in certain regimes when the corresponding matrix is numerically sparse. The proof of this appears in Appendix B.2.

**Theorem 12 (Numerically Sparse Regression Runtime)** *For solving $\epsilon$-approximate regression (Definition 2), if $\kappa \leq d^2$, SVRG framework from Theorem 14 and the variance bound from Lemma 11 gives an algorithm with running time $O\left(\left(\mathrm{nnz}(\boldsymbol{A}) + \sqrt{\kappa}\sum_{i\in[n]}\frac{\|a_i\|_2^2}{\mu}\sqrt{s_i}\right)\log\left(\frac{1}{\epsilon}\right)\right).$*

Combined with the additional acceleration framework mentione in Theorem 15, we can get an accelerated algorithm for solving regression. The proof of this appears in Appendix B.2.2.

**Theorem 13 (Numerically Sparse Accelerated Regression Runtime)** *For solving $\epsilon$-approximate regression (Definition 2) if $\kappa \leq d^2$, SVRG framework from Theorem 14, acceleration framework*

*from Theorem 15 and the variance bound from Lemma 11 gives an algorithm with running time*

$$O\left(\mathrm{nnz}(\boldsymbol{A})^{2/3}\kappa^{1/6}\left(\sum_{i\in[n]}\frac{\|a_i\|_2^2}{\mu}\sqrt{s_i}\right)^{1/3}\log(\kappa)\log\left(\frac{1}{\epsilon}\right)\right)$$

## Acknowledgments

We would like to thank the anonymous reviewers who helped improve the readability and presentation of this draft by providing many helpful comments.

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
