[Supplementary Material · supplementary.pdf]

# A  Preliminaries

## A.1  SVRG

We state the SVRG framework which was originally given by [12]. Our algorithms for solving regression and eigenvector computation both involve linear system solvers which are solved using the SVRG framework. SVRG is used to get linear convergence for stochastic gradient descent by taking gradient updates which equals the exact gradient in expectation but which have reduced variance which goes down to 0 as we reach near the optimum. This specific statement of the results is taken from [10].

**Theorem 14 (SVRG for Sums of Non-Convex functions)** *Let D be a distribution over functions,* $g_1, g_2, \ldots, g_n \in \mathbb{R}^d \to \mathbb{R}^d$. *Let* $\nabla f(x) - \nabla f(y) = \mathbb{E}_{g_k \sim D} \nabla g_k(x) - \nabla g_k(y) \quad \forall x, y \in \mathbb{R}^d$ *and let* $x^* = \operatorname{argmin}_{x \in \mathbb{R}^d} f(x)$. *Suppose that starting from some initial point* $x_0 \in \mathbb{R}^d$ *in each iteration k, we let*

$$x_{k+1} := x_k - \eta(\nabla g_i(x_k) - \nabla g_i(x_0)) + \eta \nabla f(x_0)$$

*where* $g_i \sim D$ *independently at random for some* $\eta$.

*If f is* $\mu$-*strongly convex and if for all* $x \in \mathbb{R}^d$, *we have*

$$\mathbb{E}_{g_i \sim D} \|\nabla g_i(x) - \nabla g_i(x^*)\|_2^2 \leq 2\sigma^2(f(x) - f(x^*)) \tag{1}$$

*where* $\sigma^2$ *we call the* variance parameter, *then for all* $m \geq 1$, *we have*

$$\mathbb{E}\left[ \frac{1}{m} \sum_{k \in [m]} f(x_k) - f(x^*) \right] \leq \frac{1}{1 - 2\eta\sigma^2} \left( \frac{1}{m\eta\mu} + 2\eta\sigma^2 \right) (f(x_0) - f(x^*))$$

*Consequently, if we pick* $\eta$ *to be a sufficiently small multiple of* $\frac{1}{\sigma^2}$ *then when* $m = O(\frac{\sigma^2}{\mu})$, *we can decrease the error by a constant multiplicative factor in expectation.*

The proof is taken from [10]. We are stating it here just for completeness.

**Proof** Using the fact that we have, $\nabla f(x) - \nabla f(y) = \mathbb{E}_{g_i \sim D} \nabla g_i(x) - \nabla g_i(y) \quad \forall x, y \in \mathbb{R}^d$, we have that:

$$\begin{aligned}
\mathbb{E}_{g_i \sim D} \|x_{k+1} - x^*\|_2^2 &= \mathbb{E}_{g_i \sim D} \|x_k - \eta(\nabla g_i(x_k) - \nabla g_i(x_0) + \nabla f(x_0)) - x^*\|_2^2 \\
&= \mathbb{E}_{g_i \sim D} \|x_k - x^*\|_2^2 - 2\eta \mathbb{E}_{g_i \sim D} (\nabla g_i(x_k) - \nabla g_i(x_0) + \nabla f(x_0))^\top (x_k - x^*) + \\
&\quad \eta^2 \mathbb{E}_{g_i \sim D} \|\nabla g_i(x_k) - \nabla g_i(x_0) + \nabla f(x_0)\|_2^2 \\
&= \mathbb{E}_{g_i \sim D} \|x_k - x^*\|_2^2 - 2\eta \nabla f(x_k)^\top (x_k - x^*) + \\
&\quad \eta^2 \mathbb{E}_{g_i \sim D} \|\nabla g_i(x_k) - \nabla g_i(x_0) + \nabla f(x_0)\|_2^2 \tag{2}
\end{aligned}$$

Now, using $\|x + y\|_2^2 \leq 2\|x\|_2^2 + 2\|y\|_2^2$, we get:

$$\begin{aligned}
\mathbb{E}_{g_i \sim D} \|\nabla g_i(x_k) - \nabla g_i(x_0) + \nabla f(x_0)\|_2^2 \leq 2 \mathbb{E}_{g_i \sim D} \|\nabla g_i(x_k) - \nabla g_i(x^*)\|_2^2 + \\
2 \mathbb{E}_{g_i \sim D} \|\nabla g_i(x_0) - \nabla g_i(x^*) - \nabla f(x_0)\|_2^2 \tag{3}
\end{aligned}$$

Now, we know that $\nabla f(x^*) = 0$ and using $\mathbb{E} \|x - \mathbb{E} x\|^2 \leq \mathbb{E} \|x\|_2^2$

$$\begin{aligned}
\mathbb{E}_{g_i \sim D} \|\nabla g_i(x_0) - \nabla g_i(x^*) - \nabla f(x_0)\|_2^2 &= \mathbb{E}_{g_i \sim D} \|\nabla g_i(x_0) - \nabla g_i(x^*) - (\nabla f(x_0) - \nabla f(x^*))\|_2^2 \\
&\leq \mathbb{E}_{g_i \sim D} \|\nabla g_i(x_0) - \nabla g_i(x^*)\|_2^2 \tag{4}
\end{aligned}$$

Now, using equation 1 and equation 4 in equation 3, we get:

$$\mathop{\mathbb{E}}_{g_i \sim D} \|\nabla g_i(x_k) - \nabla g_i(x_0) + \nabla f(x_0)\|_2^2 \le 4\sigma^2(f(x_k) - f(x^*)) + 4\sigma^2(f(x_0) - f(x^*))$$

$$\le 4\sigma^2(f(x_k) - f(x^*) + f(x_0) - f(x^*)) \qquad (5)$$

Using the convexity of $f$, we get $f(x^*) - f(x_k) \ge \nabla f(x_k)^\top(x^* - x^k)$, using this and equation 5 in equation 2, we get

$$\mathop{\mathbb{E}}_{g_k \sim D} \|x_{k+1} - x^*\|_2^2 \le \|x_k - x^*\|_2^2 - 2\eta \nabla f(x_k)^\top(x_k - x^*)$$
$$+ 4\eta^2\sigma^2(f(x_k) - f(x^*) + f(x_0) - f(x^*))$$
$$\le \|x_k - x^*\|_2^2 - 2\eta(f(x_k) - f(x^*))$$
$$+ 4\eta^2\sigma^2(f(x_k) - f(x^*) + f(x_0) - f(x^*))$$
$$\le \|x_k - x^*\|_2^2 - 2\eta(1 - 2\eta\sigma^2)(f(x_k) - f(x^*)) + 4\eta^2\sigma^2(f(x_0) - f(x^*))$$

Rearranging, we get that

$$2\eta(1 - 2\eta\sigma^2)(f(x_k) - f(x^*)) \le \|x_k - x^*\|_2^2 - \mathop{\mathbb{E}}_{g_k \sim D} \|x_{k+1} - x^*\|_2^2 + 4\eta^2\sigma^2(f(x_0) - f(x^*))$$

Summing over all iterations and taking expectations, we get

$$2\eta(1 - 2\eta\sigma^2)\mathbb{E}[\sum_{k \in [m]} (f(x_k) - f(x^*))] \le \|x_0 - x^*\|_2^2 + 4m\eta^2\sigma^2(f(x_0) - f(x^*))$$

Now, using strong convexity, we get that $\|x_0 - x^*\|_2^2 \le \frac{2}{\mu}(f(x_0) - f(x^*))$ and using this we get:

$$2\eta(1 - 2\eta\sigma^2)\mathbb{E}\left[\sum_{k \in [m]} (f(x_k) - f(x^*))\right] \le \frac{2}{\mu}(f(x_0) - f(x^*)) + 4m\eta^2\sigma^2(f(x_0) - f(x^*))$$

$$\mathbb{E}\left[\frac{1}{m}\sum_{k \in [m]} (f(x_k) - f(x^*))\right] \le \frac{1}{1 - 2\eta\sigma^2}\left(\frac{1}{m\eta\mu} + 2\eta\sigma^2\right)(f(x_0) - f(x^*))$$

## A.2 Acceleration

Below is a Theorem from [9] which shows how can we accelerate an ERM problem where the objective is strongly convex and each of the individual components is smooth in a black box fashion by solving many regularized version of the problems. We will use this theorem to give accelerated runtimes for our problems of regression and top eigenvector computation.

**Theorem 15 (Accelerated Approximate Proximal Point, Theorem 1.1 of [9])** *Let $f : \mathbb{R}^n \to \mathbb{R}$ be a $\mu$ strongly convex function and suppose that for all $x_0 \in \mathbb{R}^n$, $c > 0$, $\lambda > 0$, we can compute a possibly random $x_c \in \mathbb{R}^n$ such that*

$$\mathbb{E} f(x_c) - \min_{x \in \mathbb{R}^n}\left\{f(x) + \frac{\lambda}{2}\|x - x_0\|_2^2\right\} \le \frac{1}{c}\left[f(x_0) - \min_{x \in \mathbb{R}^n}\{f(x) + \frac{\lambda}{2}\|x - x_0\|_2^2\}\right]$$

*in time $T_c$. Then, given any $x_0, c > 0, \lambda \ge 2\mu$, we can compute $x_1$ such that*

$$\mathbb{E} f(x_1) - \min_x f(x) \le \frac{1}{c}[f(x_0) - \min_x f(x)]$$

*in time $O\left(T_{4\left(\frac{2\lambda+\mu}{\mu}\right)^{\frac{3}{2}}} \sqrt{\lceil\frac{\lambda}{\mu}\rceil}\log(c)\right)$*

# B Proofs

## B.1 Sampling Techniques Proofs

First, we provide the following Lemma 16 which will be used later in the proofs to relate the difference between function values at any point $x$ and the optimal point $x^*$ to the $A^\top A$ norm of difference between the two points. This is key to relating the error from sampling to function error. Note that this is standard and well known.

**Lemma 16** *Let $f(x) = \frac{1}{2}\|Ax - b\|_2^2$ and $x^* = \operatorname{argmin} f(x)$, then*

$$2(f(x) - f(x^*)) = \|A(x - x^*)\|_2^2$$

**Proof** We know $\nabla f(x^*) = 0$ since $x^* = \operatorname{argmin} f(x)$, thus, we get that $A^\top(Ax^* - b) = 0$ or $A^\top Ax^* = A^\top b$. Now,

$$
\begin{aligned}
2(f(x) - f(x^*)) &= \|Ax - b\|_2^2 - \|Ax^* - b\|_2^2 \\
&= (Ax - b - Ax^* + b)^\top (Ax - 2b + Ax^*) \\
&= (x - x^*)^\top A^\top (Ax - 2b + Ax^*) \\
&= (x - x^*)^\top (A^\top Ax - 2A^\top b + A^\top Ax^*) \\
&= (x - x^*)^\top (A^\top Ax - 2A^\top Ax^* + A^\top Ax^*) \\
&= (x - x^*)^\top A^\top A(x - x^*) \\
&= \|A(x - x^*)\|_2^2
\end{aligned}
$$

**Lemma 3 (Numerical Sparsity)** *For $a \in \mathbb{R}^d$ and $c \in [d]$, we have $\|\bar{\Pi}_c(a)\|_2^2 \le s(a)\|a\|_2^2/c$.*

**Proof** We can assume without loss of generality that $|a_i| \ge |a_j|$ whenever $i < j$ i.e. the indices are sorted in descending order of the absolute values.

$$
\begin{aligned}
\frac{\|\bar{\Pi}_c(a)\|_2^2}{\|a\|_2^2} &= \frac{a_{c+1}^2 + a_{c+2}^2 + \cdots + a_d^2}{\|a\|_2^2} \le \frac{|a_{c+1}|(|a_{c+1}| + |a_{c+2}| + \cdots + |a_d|)}{\|a\|_2^2} \\
&\le \frac{|a_{c+1}|\|a\|_1}{\|a\|_2^2} \le \frac{\|a\|_1\|a\|_1}{c\|a\|_2^2} \le \frac{\|a\|_1^2}{c\|a\|_2^2} \le \frac{s(a)}{c} \ .
\end{aligned}
$$

**Lemma 4 (Stochastic Approximation of $a$)** *Let $a \in \mathbb{R}^d$ and $c \in \mathbb{N}$ and let our estimator $(\hat{a})_c = \texttt{Samplevec}(a, x)$ (Algorithm 1) Then,*

$$\mathbb{E}[(\hat{a})_c] = a \ \ and \ \ \mathbb{E}\left[\|(\hat{a})_c\|_2^2\right] \le \|a\|_2^2 \left(1 + \frac{s(a)}{c}\right)$$

**Proof** Since, $(\hat{a})_c = \texttt{Samplevec}(a, c)$, we can also write this as $(\hat{a})_c = \frac{1}{c}\sum_{i=1}^c X_i$ where $\{X_i\}$ are sampled i.i.d. such that $\Pr(X_i = \frac{a_j}{p_j}\hat{e}_j) = p_j = \frac{|a_j|}{\|a\|_1} \quad \forall j \in [d]$.

Calculating first and second moments of random variable $X_i$, we get that

$$
\begin{aligned}
\mathbb{E}[X_i] &= \sum_{j \in d} p_j \frac{a_j}{p_j}\hat{e}_j = \sum_{j \in d} \hat{e}_j a_j \\
&= a
\end{aligned}
\tag{6}
$$

$$
\begin{aligned}
\mathbb{E}\left[\|X_i\|_2^2\right] &= \sum_{j \in [d]} p_j \left(\frac{a_j \hat{e}_j}{p_j}\right)^2 = \sum_{j \in [d]} \frac{a_j^2}{p_j} = \|a\|_1 \sum_{j \in [d]} \frac{a_j^2}{|a_j|} = \|a\|_1 \sum_{j \in [d]} |a_j| \\
&= \|a\|_1^2
\end{aligned}
\tag{7}
$$

Now, using the calculated moments in equation 6 and equation 7, to calculate the first and second moments of $(\hat{a})_c$

$$\mathbb{E}[(\hat{a})_c] = \mathbb{E}\left[\frac{1}{c}\sum_{i\in[c]}X_i\right] = \frac{1}{c}\sum_{i\in[c]}\mathbb{E}[X_i] = \frac{1}{c}\sum_{i\in[c]}a = a$$

$$\mathbb{E}\left[\|(\hat{a})_c\|_2^2\right] = \mathbb{E}\left[\left\|\frac{1}{c}\sum_{i\in[c]}X_i\right\|_2^2\right]$$

$$= \frac{1}{c^2}\mathbb{E}\left[\sum_{i\in[c]}\|X_i\|_2^2 + \sum_{i,j\in[c],i\neq j}X_i^\top X_j\right]$$

Using the moments for random variable $X_i$ calculated in equation 6 and equation 7 and independence of $X_i$ and $X_j$ for $i\neq j$, we get that

$$\mathbb{E}\left[\|(\hat{a})_c\|_2^2\right] = \frac{1}{c^2}\left(\sum_{i\in[c]}\|a\|_1^2 + \sum_{i,j\in[c],i\neq j}a^\top a\right)$$

$$= \frac{1}{c^2}\left(c\|a\|_1^2 + c(c-1)a^\top a\right)$$

$$= \frac{1}{c}\left(\|a\|_1^2 + (c-1)a^\top a\right)$$

Using $s(a) = \|a\|_1^2/\|a\|_2^2$

$$\mathbb{E}[\|(\hat{a})_c\|_2^2] = \|a\|_2^2\frac{1}{c}\left(s(a) + (c-1)\right) \leq \|a\|_2^2\left(1 + \frac{s(a)}{c}\right)$$

**Lemma 5 (Stochastic Approximation of $a^\top x$)** *Let $a, x \in \mathbb{R}^d$ and $c \in [d]$, and let our estimator be defined as $\widehat{(a^\top x)}_c = \mathtt{Sampledotproduct}(a, x, c)$ (Algorithm 2) Then,*

$$\mathbb{E}[\widehat{(a^\top x)}_c] = a^\top x \ \ and \ \ \mathbb{E}\left[\widehat{(a^\top x)}_c^2\right] \leq (a^\top x)^2 + \frac{1}{c}\|\bar{\Pi}_c(a)\|_2^2\|x\|_2^2$$

**Proof** Since $\widehat{(a^\top x)}_c = \mathtt{Sampledotproduct}(a, x, c)$, we can also write this as $\widehat{(a^\top x)}_c = \Pi_c(a)^\top x + \frac{1}{c}\sum_{i=1}^c X_i(x)$ where $\{X_i\}$ are sampled i.i.d. such that for each $X_i$, $Pr(X_i = \frac{a_k x_k}{p_k}) = p_k = \frac{a_k^2}{\|\bar{\Pi}_c(a)\|_2^2} \ \ \forall k \in \bar{I}_c(a)$.

Calculating first and second moments of random variable $X_i$, we get that

$$\mathbb{E}[X_i] = \sum_{k\in\bar{I}_c(a_i)}p_k\frac{a_k x_k}{p_k} = \sum_{k\in\bar{I}_c(a_i)}a_k x_k$$

$$= \bar{\Pi}_c(a)^\top x \tag{8}$$

$$\mathbb{E}\left[\|X_i\|_2^2\right] = \sum_{k\in\bar{I}_c(a_i)}p_k\left(\frac{a_k x_k}{p_k}\right)^2 = \sum_{k\in\bar{I}_c(a)}\frac{a_k^2 x_k^2}{p_k} = \sum_{k\in\bar{I}_c(a)}\frac{\|\bar{\Pi}_c(a)\|_2^2 a_k^2 x_k^2}{a_k^2} = \sum_{k\in\bar{I}_c(a)}\|\bar{\Pi}_c(a)\|_2^2 x_k^2$$

$$\leq \|\bar{\Pi}_c(a)\|_2^2\|x\|_2^2 \tag{9}$$

Using the moments calculated in equation 8 and equation 9, we calculate the first and second moments of the estimator $\widehat{(a^\top x)}_c$

$$
\begin{aligned}
\mathbb{E}[\widehat{(a^\top x)}_c] &= \mathbb{E}\left[\Pi_c(a)^\top x + \frac{1}{c}\sum_{i\in[c]} X_i\right] \\
&= \Pi_c(a)^\top x + \frac{1}{c}\sum_{i\in[c]} \mathbb{E}[X_i] \\
&= \Pi_c(a)^\top x + \frac{1}{c}\sum_{i\in[c]} \bar{\Pi}_c(a)^\top x \\
&= \Pi_c(a)^\top x + \bar{\Pi}_c(a)^\top x \\
&= a^\top x
\end{aligned}
$$

$$
\begin{aligned}
\mathbb{E}\left[\widehat{(a^\top x)}_c^2\right] &= \mathbb{E}\left[\left(\Pi_c(a)^\top x + \frac{1}{c}\sum_{i\in[c]} X_i\right)^2\right] \\
&= \mathbb{E}\left[(\Pi_c(a)^\top x)^2 + 2\Pi_c(a)^\top x \frac{1}{c}\sum_{i\in[c]} X_i + \frac{1}{c^2}\left(\sum_{i\in[c]} X_i\right)^2\right] \\
&= \mathbb{E}\left[(\Pi_c(a)^\top x)^2\right] + \mathbb{E}\left[2\Pi_c(a)^\top x \frac{1}{c}\sum_{i\in[c]} X_i\right] + \mathbb{E}\left[\frac{1}{c^2}\left(\sum_{i\in[c]} X_i\right)^2\right] \\
&= (\Pi_c(a)^\top x)^2 + 2\Pi_c(a)^\top x \frac{1}{c}\sum_{i\in[c]} \mathbb{E}[X_i] + \mathbb{E}\left[\frac{1}{c^2}\left(\sum_{i\in[c]} X_i\right)^2\right]
\end{aligned}
$$

Using the expectation of the random variable $X_i$, calculated in equation 8

$$
\begin{aligned}
\mathbb{E}\left[\widehat{(a^\top x)}_c^2\right] &= (\Pi_c(a)^\top x)^2 + 2\Pi_c(a)^\top x \frac{1}{c}\sum_{i\in[c]} \bar{\Pi}_c(a)^\top x + \mathbb{E}\left[\frac{1}{c^2}\left(\sum_{i\in[c]} X_i\right)^2\right] \\
&= (\Pi_c(a)^\top x)^2 + 2\Pi_c(a)^\top x \bar{\Pi}_c(a)^\top x + \frac{1}{c^2}\mathbb{E}\left[\sum_{i\in[c]} X_i^2 + \sum_{i,j\in[c],i\neq j} X_i \cdot X_j\right]
\end{aligned}
$$

Using the independence of $X_i$ and $X_j$, we get that

$$
\mathbb{E}[\widehat{(a^\top x)}_c^2] = (\Pi_c(a)^\top x)^2 + 2\Pi_c(a)^\top x \bar{\Pi}_c(a)^\top x + \frac{1}{c^2}\left(\sum_{i\in[c]} \mathbb{E}\left[X_i^2\right] + \sum_{i,j\in[c],i\neq j} \mathbb{E}[X_i]\cdot\mathbb{E}[X_j]\right)
$$

Using the first and second moments of the random variable $X_i$, calculated in equation 8 and equation 9

$$\mathbb{E}[(\widehat{a^\top x})_c^2] = (\Pi_c(a)^\top x)^2 + 2\Pi_c(a)^\top x \bar{\Pi}_c(a)^\top x$$

$$+ \frac{1}{c^2}\left(\sum_{i\in[c]}\|\bar{\Pi}_c(a)\|_2^2\|x\|_2^2 + \sum_{i,j\in[c],i\neq j}\bar{\Pi}_c(a)^\top x \bar{\Pi}_c(a)^\top x\right)$$

$$= (\Pi_c(a)^\top x)^2 + 2\Pi_c(a)^\top x \bar{\Pi}_c(a)^\top x$$

$$+ \frac{1}{c^2}\left(c\|\bar{\Pi}_c(a)\|_2^2\|x\|_2^2 + c(c-1)\bar{\Pi}_c(a)^\top x \bar{\Pi}_c(a)^\top x\right)$$

$$= (\Pi_c(a)^\top x)^2 + 2\Pi_c(a)^\top x \bar{\Pi}_c(a)^\top x + \frac{1}{c}\|\bar{\Pi}_c(a)\|_2^2\|x\|_2^2 + \left(1 - \frac{1}{c}\right)\bar{\Pi}_c(a)^\top x \bar{\Pi}_c(a)^\top x$$

$$\leq (\Pi_c(a)^\top x)^2 + 2\Pi_c(a)^\top x \bar{\Pi}_c(a)^\top x + \frac{1}{c}\|\bar{\Pi}_c(a)\|_2^2\|x\|_2^2 + \bar{\Pi}_c(a)^\top x \bar{\Pi}_c(a)^\top x$$

Using $(\Pi_c(a)^\top x)^2 + 2\Pi_c(a)^\top x \bar{\Pi}_c(a)^\top x + \bar{\Pi}_c(a)^\top x \bar{\Pi}_c(a)^\top x = (a^\top x)^2$, we get that

$$\mathbb{E}[(\widehat{a^\top x})_c^2] \leq (a^\top x)^2 + \frac{1}{c}\|\bar{\Pi}_c(a)\|_2^2\|x\|_2^2$$

**Lemma 6 (Stochastic Approximation of $aa^\top x$)** *Let $a, x \in \mathbb{R}^d$ and $c \in [d]$, and the estimator be defined as $(\widehat{aa^\top x})_c = \texttt{Samplerankonemat}(a, x, c)$ (Algorithm 3) Then,*

$$\mathbb{E}[(\widehat{aa^\top x})_c] = aa^\top x \ \text{ and } \ \mathbb{E}\left[\|(\widehat{aa^\top x})_c\|_2^2\right] \leq \|a\|_2^2\left(1 + \frac{s(a)}{c}\right)\left((a^\top x)^2 + \frac{s(a)}{c^2}\|a\|_2^2\|x\|_2^2\right)$$

**Proof** Since, $(\widehat{aa^\top x})_c = (\widehat{a})_c (\widehat{a^\top x})_c$ where $(\widehat{a})_c, (\widehat{a^\top x})_c$ are the estimators for $a$ and $a^\top x$ defined in Lemma 4 and Lemma 5 respectively and formed using independent samples. First calculating the expectation of $(\widehat{aa^\top x})_c$

$$\mathbb{E}[(\widehat{aa^\top x})_c] = \mathbb{E}[(\widehat{a})_c(\widehat{a^\top x})_c] = \mathbb{E}[(\widehat{a})_c]\,\mathbb{E}[(\widehat{a^\top x})_c] = aa^\top x$$

The above proof uses the fact that $(\widehat{a})_c$ and $(\widehat{a^\top x})_c$ are estimated using independent samples. Now, calculating the second moment of $\|(\widehat{aa^\top x})_c\|_2$, we get that

$$\mathbb{E}\left[\|(\widehat{aa^\top x})_c\|_2^2\right] = \mathbb{E}\left[\|(\widehat{a})_c(\widehat{a^\top x})_c\|_2^2\right]$$

$$= \mathbb{E}\left[\|(\widehat{a})_c\|_2^2\|(\widehat{a^\top x})_c\|_2^2\right]$$

$$= \mathbb{E}\left[\|(\widehat{a})_c\|_2^2\right]\mathbb{E}\left[(\widehat{aa^\top x})_c^2\right]$$

$$\leq \|a\|_2^2\left(1 + \frac{s(a)}{c}\right)\left((a^\top x)^2 + \frac{1}{c}\|\bar{\Pi}_c(a)\|_2^2\|x\|_2^2\right)$$

Now, using Lemma 3, we know that $\|\bar{\Pi}_c(a)\|_2^2 \leq \frac{s(a)}{c}\|a\|_2^2$
Thus, we get that

$$\mathbb{E}\left[\|(\widehat{aa^\top x})_c\|_2^2\right] \leq \|a\|_2^2\left(1 + \frac{s(a)}{c}\right)\left((a^\top x)^2 + \frac{s(a)}{c^2}\|a\|_2^2\|x\|_2^2\right)$$

**Lemma 7 (Stochastic Approximation of $A^\top Ax$ )** *Let $A \in \mathbb{R}^{n\times d}$ with rows $a_1, a_2, \ldots, a_n$ and $x \in \mathbb{R}^d$ and let $(\widehat{A^\top Ax})_k = \texttt{Samplemat}(A, x, k)$ (Algorithm 4) where $k$ is some parameter. Then,*

$$\mathbb{E}\left[(\widehat{A^\top Ax})_k\right] = A^\top Ax \ \text{ and } \ \mathbb{E}\left[\left\|(\widehat{A^\top Ax})_k\right\|_2^2\right] \leq M\left(\|Ax\|_2^2 + \frac{1}{k^2}\|A\|_F^2\|x\|_2^2\right).$$

**Proof** Since, $(\widehat{A^\top Ax})_k = \frac{1}{p_i}(\widehat{a_i a_i^\top x})_{c_i}$ with probability $p_i = \frac{\|a_i\|_2^2}{M}\left(1 + \frac{s_i}{c_i}\right)$ where $M$ is the normalization constant where $(\widehat{a_i a_i^\top x})_{c_i}$ is the estimator of $a_i a_i^\top x$ defined in Lemma 6 and are formed independently of each other and independently of $i$ where $s_i = s(a_i)$ and $k$ is some parameter such that $c_i = \sqrt{s_i}k \le d$. Calculating the expectation of $(\widehat{A^\top Ax})_k$, we get

$$\mathbb{E}\left[(\widehat{A^\top Ax})_k\right] = \mathbb{E}\left[\frac{1}{p_i}\mathbb{E}[(\widehat{a_i a_i^\top x})_{c_i}]\right] = \mathbb{E}\left[\frac{1}{p_i}a_i a_i^\top x\right] = \sum_i \frac{p_i}{p_i} a_i a_i^\top x = \sum_i a_i a_i^\top x = A^\top Ax$$

In the proof above, we used the expectation of $(\widehat{a_i a_i^\top x})_{c_i}$ calculated in Lemma 6 and also used that $i$ and $(\widehat{a_i a_i^\top x})_{c_i}$ are chosen independently of each other.

Calculating the second moment of $(\widehat{A^\top Ax})_k$, we get

$$\mathbb{E}\left[\left\|(\widehat{A^\top Ax})_k\right\|_2^2\right] = \mathbb{E}\left[\left\|\frac{1}{p_i}(\widehat{a_i a_i^\top x})_{c_i}\right\|_2^2\right] = \mathbb{E}\left[\frac{1}{p_i^2}\mathbb{E}\left[\|(\widehat{a_i a_i^\top x})_{c_i}\|_2^2\right]\right]$$

$$\le \mathbb{E}\left[\frac{1}{p_i^2}\|a_i\|_2^2\left(1 + \frac{s_i}{c_i}\right)\left((a_i^\top x)^2 + \frac{s_i}{c_i^2}\|a_i\|_2^2\|x\|_2^2\right)\right]$$

$$\le \sum_i \frac{1}{p_i}\|a_i\|_2^2\left(1 + \frac{s_i}{c_i}\right)\left((a_i^\top x)^2 + \frac{s_i}{c_i^2}\|a_i\|_2^2\|x\|_2^2\right)$$

Putting the value of $p_i = \frac{\|a_i\|_2^2}{M}\left(1 + \frac{s_i}{c_i}\right)$, we get

$$\mathbb{E}\left[\left\|(\widehat{A^\top Ax})_k\right\|_2^2\right] \le M\sum_i\left((a_i^\top x)^2 + \frac{s_i}{c_i^2}\|a_i\|_2^2\|x\|_2^2\right)$$

Now, putting the value of $c_i = \sqrt{s_i}k$, we get

$$\mathbb{E}\left[\left\|(\widehat{A^\top Ax})_k\right\|_2^2\right] \le M\sum_i\left((a_i^\top x)^2 + \frac{1}{k^2}\|a_i\|_2^2\|x\|_2^2\right)$$

$$\le M\left(\|Ax\|_2^2 + \frac{1}{k^2}\|A\|_F^2\|x\|_2^2\right)$$

## B.2 Application Proofs

### B.2.1 Top Eigenvector Computation

**Lemma 8 (Variance bound for eigenvector computation)** *Let* $\nabla g(x) = \lambda x - (\widehat{A^\top Ax})_k$ *where* $(\widehat{A^\top Ax})_k$ *is the estimator of* $A^\top Ax$ *defined in Lemma 7, and* $k = \sqrt{\mathrm{sr}(A)}$, *then we get*

$$\mathbb{E}[\nabla g(x)] = (\lambda I - A^\top A)x \quad and \quad \mathbb{E}\left[\|\nabla g(x) - \nabla g(x^*)\|_2^2\right] \le (f(x) - f(x^*))8M/\mathrm{gap}$$

*with average time taken in calculating* $\nabla g(x), T = \sum_i \|a_i\|_2^2\left(s_i + \sqrt{s_i \mathrm{sr}(A)}\right)/M$ *where* $M = \sum_i \|a_i\|_2^2\left(1 + \sqrt{\frac{s_i}{\mathrm{sr}(A)}}\right)$ *and* $f(x) = \frac{1}{2}x^\top Bx - b^\top x$

**Proof** Using the unbiasedness of the estimator $(\widehat{A^\top Ax})_k$ from Lemma 7, we get $\mathbb{E}[\nabla g(x)] = \mathbb{E}[\lambda x - (\widehat{A^\top Ax})_k] = \lambda x - A^\top Ax$

Putting the second moment of $(\widehat{A^\top Ax})_k$ from Lemma 7 and since in one iteration, the same $\nabla g(x)$

is picked, the randomness in $\nabla g(x)$ and $\nabla g(x^*)$ is same, therefore, we get

$$\mathbb{E}\left[\|\nabla g(x) - \nabla g(x^*)\|_2^2\right] = \mathbb{E}\left[\|\lambda(x - x^*) - (\boldsymbol{A}^\top \widehat{\boldsymbol{A}(x - x^*)})_k\|_2^2\right]$$

$$= \lambda^2\|x - x^*\|_2^2 - 2\lambda(x - x^*)^\top \mathbb{E}\left[(\boldsymbol{A}^\top \widehat{\boldsymbol{A}(x - x^*)})_k\right]$$

$$+ \mathbb{E}\left[\|(\boldsymbol{A}^\top \widehat{\boldsymbol{A}(x - x^*)})_k\|_2^2\right]$$

Using the unbiasedness and second moment bound of $(\widehat{\boldsymbol{A}^\top \boldsymbol{A}x})_k$ from Lemma 7, we get

$$\mathbb{E}\left[\|\nabla g(x) - \nabla g(x^*)\|_2^2\right] = \lambda^2(x - x^*)^2 - 2\lambda\|\boldsymbol{A}(x - x^*)\|_2^2$$

$$+ M\left(\|\boldsymbol{A}(x - x^*)\|_2^2 + \frac{1}{k^2}\|\boldsymbol{A}\|_F^2\|x - x^*\|_2^2\right)$$

$$\leq \lambda^2(x - x^*)^2 + M\left(\|\boldsymbol{A}(x - x^*)\|_2^2 + \frac{1}{k^2}\|\boldsymbol{A}\|_F^2\|x - x^*\|_2^2\right)$$

Now, since we want to relate the variance of our gradient estimator the function error to be used in the SVRG framework, using the strong convexity parameter of the matrix $\boldsymbol{B}$, we get the following:

$$\mathbb{E}\left[\|\nabla g(x) - \nabla g(x^*)\|_2^2\right] \leq \|x - x^*\|_{\boldsymbol{B}}^2 \frac{\lambda^2}{\lambda - \lambda_1} + M\|x - x^*\|_{\boldsymbol{B}}^2\left(\frac{\lambda_1}{\lambda - \lambda_1} + \frac{\|\boldsymbol{A}\|_F^2}{(\lambda - \lambda_1)k^2}\right)$$

Now, using $k^2 = \text{sr}(\boldsymbol{A})$ and rewriting the equation in terms of problem parameters, $\text{sr}(\boldsymbol{A}) = \frac{\|\boldsymbol{A}\|_F^2}{\lambda_1}$ and $\lambda = (1 + c\text{gap})\lambda_1$ where $c$ is some constant and using $\text{gap} \leq 1$ we get

$$\mathbb{E}\left[\|\nabla g(x) - \nabla g(x^*)\|_2^2\right] \leq \|x - x^*\|_{\boldsymbol{B}}^2 \frac{4\lambda_1}{\text{gap}} + \frac{2\|x - x^*\|_{\boldsymbol{B}}^2 M}{\text{gap}} \leq 2(f(x) - f(x^*))\left(\frac{4\lambda_1}{\text{gap}} + \frac{2M}{\text{gap}}\right)$$

It is easy to see that $\lambda_1 \leq M$ and hence, the second term always upper bounds the first term, thus we get the desired variance bound. Note from Theorem 14, we know that the gradient update is of the following form.

$$x_{k+1} := x_k - \eta(\nabla g_i(x_k) - \nabla g_i(x_0)) + \eta\nabla f(x_0).$$

Note, the estimator $(\widehat{\boldsymbol{A}^\top \boldsymbol{A}x})_k$ uses $(\hat{a})_{c_i}$ and $(\widehat{a^\top x})_{c_i}$ estimators internally which both use $c_i = \sqrt{s_i}k$ coordinates. Also, $\lambda x_k$ can be added to $x_k$ in $O(1)$ time by just maintaining a multiplicative coefficient of the current iterate and doing the updates accordingly. Hence, our estimator of $\nabla g_i(x_k) - \nabla g_i(x_0)$ can be implemented in $O(c_i)$ time when the ith row is chosen. Furthermore, the dense part $\nabla f(x_0) = \boldsymbol{B}x_0 - b$ can be added in $O(1)$ time by separately maintaining the coefficient of this fixed vector in each $x_k$ and using it as necessary to calculate the $O(c_i)$ coordinates during each iteration. Consequently, we can bound the total expected time for implementing the iterations by

$$T = \sum_i p_i c_i = \sum_i \frac{\|a_i\|_2^2}{M}\left(1 + \frac{s_i}{c_i}\right)c_i = \sum_i \frac{\|a_i\|_2^2}{M}(c_i + s_i) = \sum_i \frac{\|a_i\|_2^2}{M}\left(\sqrt{s_i\text{sr}(\boldsymbol{A})} + s_i\right)$$

**Theorem 17 (Theorem 16 of [10])** *Let us say we have a linear system solver that finds $x$ such that:*

$$\mathbb{E}\|x - x^*\|_{\boldsymbol{B}}^2 \leq \frac{1}{2}\|x_0 - x^*\|_{\boldsymbol{B}}^2$$

*in time $O(T)$ where $f(x) = \frac{1}{2}x^\top \boldsymbol{B}x - b^\top x$, $\boldsymbol{B} = \lambda \boldsymbol{I} - \boldsymbol{A}^\top \boldsymbol{A}$ and $x_0$ is some initial point. Then we can find an $\epsilon$-approximation $v$ to the top eigenvector of $\boldsymbol{A}^\top \boldsymbol{A}$ in time $O\left(T \cdot \left(\log^2\left(\frac{d}{\text{gap}}\right) + \log\left(\frac{1}{\epsilon}\right)\right)\right)$ where $\left(1 + \frac{\text{gap}}{150}\right)\lambda_1 \leq \lambda \leq \left(1 + \frac{\text{gap}}{100}\right)\lambda_1$*

Theorem 16 of [10] states the running time in terms of the running time required for their system solver but it can be replaced with any other $\epsilon$-approximate linear system solver.

**Theorem 18 (Linear System Solver Runtime for $\boldsymbol{B} = \lambda \boldsymbol{I} - \boldsymbol{A}^\top \boldsymbol{A}$)** *For a matrix $\boldsymbol{B} = \lambda \boldsymbol{I} - \boldsymbol{A}^\top \boldsymbol{A}$, we have an algorithm which returns $x$ such that $\mathbb{E}\|x - x^*\|_{\boldsymbol{B}}^2 \leq \frac{1}{2}\|x^0 - x^*\|_{\boldsymbol{B}}^2$ in running time*

$$O\left(\mathrm{nnz}(\boldsymbol{A}) + \frac{1}{\mathrm{gap}^2 \lambda_1} \sum_i \|a_i\|_2^2 (\sqrt{s_i} + \sqrt{\mathrm{sr}(\boldsymbol{A})})\sqrt{s_i}\right)$$

*assuming $\lambda_1(1 + \frac{\mathrm{gap}}{150}) \leq \lambda \leq \lambda_1(1 + \frac{\mathrm{gap}}{100})$ where $x^* = \mathrm{argmin}_{x \in \mathbb{R}^d} \frac{1}{2} x^\top \boldsymbol{B} x - c^\top x$*

**Proof** The problem of solving $\min f(x)$ where $f(x) = \frac{1}{2} x^\top \boldsymbol{B} x - c^\top x$ can be solved by using the SVRG framework defined in Theorem 14 with the strong convexity parameter $\mu = \lambda - \lambda_1(\boldsymbol{A}^\top \boldsymbol{A})$. Using the estimator for $\nabla f(x)$ defined in Lemma 8, we get the corresponding variance defined in Theorem 14 as $\frac{4M}{\mathrm{gap}}$ i.e. $\sigma^2 = \frac{4M}{\mathrm{gap}}$ where $M$ is as defined in Lemma 8. Therefore, according to Theorem 14, we can decrease the error by a constant factor in total number of iterations $O(\frac{4M}{\mathrm{gap}(\lambda - \lambda_1)})$. The expected time taken per iteration is $T = \sum_i \frac{\|a_i\|_2^2}{M}\left(s_i + \sqrt{\mathrm{sr}(\boldsymbol{A})s_i}\right)$ as defined in Lemma 8. Now, again we can argue that the total time taken would be $T\sigma^2/\lambda - \lambda_1$ with constant probability by using Markov inequality to argue that time taken per iteration and number of iterations do not exceed their expected values with constant probability upto constant factors. Therefore, the total time taken to decrease the error by a constant factor would be $O\left(\mathrm{nnz}(\boldsymbol{A}) + \frac{1}{\lambda - \lambda_1} \frac{M}{\mathrm{gap}} \sum_i \frac{\|a_i\|_2^2}{M}\left(s_i + \sqrt{s_i \mathrm{sr}(\boldsymbol{A})}\right)\right)$. Simplifying, we get

$$\frac{1}{\lambda - \lambda_1} \frac{M}{\mathrm{gap}} \sum_i \frac{\|a_i\|_2^2}{M}\left(s_i + \sqrt{s_i sr(\boldsymbol{A})}\right) = \frac{1}{\mathrm{gap}^2} \sum_i \frac{\|a_i\|_2^2}{\lambda_1}\left(\sqrt{s_i} + \sqrt{sr(\boldsymbol{A})}\right)\sqrt{s_i}$$

**Theorem 9 (Numerically Sparse Top Eigenvector Computation Runtime)** *Linear system solver from Theorem 18 combined with the shift and invert framework from [10] stated in Theorem 17 gives an algorithm which computes $\epsilon$-approximate top eigenvector (Definition 1) in total running time* $O\left(\left(\mathrm{nnz}(\boldsymbol{A}) + \frac{1}{\mathrm{gap}^2 \lambda_1} \sum_i \|a_i\|_2^2 \left(\sqrt{s_i} + \sqrt{\mathrm{sr}(\boldsymbol{A})}\right)\sqrt{s_i}\right) \cdot \left(\log^2\left(\frac{d}{\mathrm{gap}}\right) + \log\left(\frac{1}{\epsilon}\right)\right)\right)$

**Proof** We get this from combining Theorem 17 and Theorem 18.

**Theorem 10 (Numerically Sparse Accelerated Top Eigenvector Computation Runtime)**
*Linear system solver from Theorem 18 combined with acceleration framework from [9] mentioned in Theorem 15 and shift and invert framework from [10] stated in Theorem 17 gives an algorithm which computes $\epsilon$-approximate top eigenvector (Definition 1) in total running time* $\tilde{O}\left(\mathrm{nnz}(\boldsymbol{A}) + \frac{\mathrm{nnz}(\boldsymbol{A})^{3/4}}{\sqrt{\mathrm{gap}}}\left(\sum_i \frac{\|a_i\|_2^2}{\lambda_1}\left(\sqrt{s_i} + \sqrt{\mathrm{sr}(\boldsymbol{A})}\right)\sqrt{s_i}\right)^{1/4}\right)$ *where $\tilde{O}$ hides a factor of* $\left(\log^2\left(\frac{d}{\mathrm{gap}}\right) + \log\left(\frac{1}{\epsilon}\right)\right)\log\left(\frac{d}{\mathrm{gap}}\right)$.

**Proof** When solving the regularized linear system in $\boldsymbol{B} + \gamma \boldsymbol{I} = (\lambda + \gamma)I - \boldsymbol{A}^\top \boldsymbol{A}$ upto constant accuracy, we get a total running time of $O\left(\mathrm{nnz}(\boldsymbol{A}) + \frac{2\lambda_1(\boldsymbol{A}^\top \boldsymbol{A})}{(\lambda + \gamma - \lambda_1)^2} \sum_i \|a_i\|_2^2 \left(\sqrt{s_i} + \sqrt{\mathrm{sr}(\boldsymbol{A})}\right)\sqrt{s_i}\right)$ by Theorem 18. Hence, the total running time for solving the unregularized linear system in $\boldsymbol{B}$ according to Theorem 15 will be
$O\left(\left(\mathrm{nnz}(\boldsymbol{A}) + \frac{2\lambda_1}{(\lambda + \gamma - \lambda_1)^2} \sum_i \|a_i\|_2^2 \left(\sqrt{s_i} + \sqrt{\mathrm{sr}(\boldsymbol{A})}\right)\sqrt{s_i}\right) \log\left(\frac{2\gamma}{\lambda - \lambda_1}\right)\sqrt{\frac{\gamma}{\lambda - \lambda_1}}\right)$ assuming $\gamma \geq 2(\lambda - \lambda_1)$ by the assumption of the theorem.

Balancing the two terms, we get that

$$\gamma = \sqrt{\frac{2\lambda_1}{\mathrm{nnz}(\boldsymbol{A})} \sum_i \|a_i\|_2^2 \left(\sqrt{s_i} + \sqrt{\mathrm{sr}(\boldsymbol{A})}\right)\sqrt{s_i}}$$

Putting this in the total runtime and using $\frac{\gamma}{\lambda - \lambda_1} \leq \sqrt{\frac{d\|\boldsymbol{A}\|_F^2}{\lambda_1 \mathrm{nnz}(\boldsymbol{A})\mathrm{gap}^2}}$, we get a total runtime of

$$\tilde{O}\left(\mathrm{nnz}(\boldsymbol{A}) + \mathrm{nnz}(\boldsymbol{A})\left(\frac{\lambda_1}{\mathrm{nnz}(\boldsymbol{A})(\lambda - \lambda_1)^2} \sum_i \|a_i\|_2^2 \left(\sqrt{s_i} + \sqrt{\mathrm{sr}(\boldsymbol{A})}\right)\sqrt{s_i}\right)^{\frac{1}{4}}\right)$$

where $\tilde{O}$ hides a factor of $\log\left(\frac{d\|\boldsymbol{A}\|_F^2}{\lambda_1 \mathrm{nnz}(\boldsymbol{A})\mathrm{gap}^2}\right)$.

Since $\frac{\|\boldsymbol{A}\|_F^2}{\lambda_1} \leq d$ and $\mathrm{nnz}(\boldsymbol{A}) \geq 1$, we get a running time of

$$O\left(\left(\mathrm{nnz}(\boldsymbol{A}) + \frac{\mathrm{nnz}(\boldsymbol{A})^{\frac{3}{4}}}{\sqrt{\mathrm{gap}}}\left(\sum_i \frac{\|a_i\|_2^2}{\lambda_1}\left(\sqrt{s_i} + \sqrt{\mathrm{sr}(\boldsymbol{A})}\right)\sqrt{s_i}\right)^{\frac{1}{4}}\right)\log\left(\frac{d}{\mathrm{gap}}\right)\right)$$

for solving a linear system in $B$. Then we get the final running time from combining Theorem 17 along with the time for the linear system solver in $\boldsymbol{B}$ obtained above.

### B.2.2 Regression

**Lemma 11 (Variance Bound for Regression)** *Let* $\nabla g(x) = (\widehat{\boldsymbol{A}^\top \boldsymbol{A}x})_k$ *where* $(\widehat{\boldsymbol{A}^\top \boldsymbol{A}x})_k$ *is the estimator for* $\boldsymbol{A}^\top \boldsymbol{A}x$ *defined in Lemma 7 and* $k = \sqrt{\kappa}$, *assuming* $\kappa \leq d^2$ *we get*

$$\mathbb{E}[\nabla g(x)] = \boldsymbol{A}^\top \boldsymbol{A}x \ \ and \ \ \mathbb{E}\left[\|\nabla g(x) - \nabla g(x^*)\|_2^2\right] \leq M(f(x) - f(x^*))$$

*with average time taken in calculating* $\nabla g(x)$, $T = \frac{\sqrt{\kappa}}{M}\sum_{i\in[n]}\|a_i\|_2^2\sqrt{s_i}$ *where* $M = \sum_i \|a_i\|_2^2\left(1 + \sqrt{\frac{s_i}{\kappa}}\right)$ *where* $f(x) = \frac{1}{2}\|\boldsymbol{A}x - b\|_2^2$

**Proof** Since we know $(\widehat{\boldsymbol{A}^\top \boldsymbol{A}x})_k$ is an unbiased estimate from Lemma 7, we get $\mathbb{E}[\nabla g(x)] = \mathbb{E}[(\widehat{\boldsymbol{A}^\top \boldsymbol{A}x})_k] = \boldsymbol{A}^\top \boldsymbol{A}x$

To calculate $\mathbb{E}\left[\|\nabla g(x) - \nabla g(x^*)\|_2^2\right]$, using the second moment of $(\widehat{\boldsymbol{A}^\top \boldsymbol{A}x})_k$ from Lemma 7 and since in one iteration, the same $\nabla g(x)$ is picked, the randomness in $\nabla g(x)$ and $\nabla g(x^*)$ is same, we get

$$\mathbb{E}\left[\|\nabla g(x) - \nabla g(x^*)\|_2^2\right] = \mathbb{E}\left[\|(\boldsymbol{A}^\top \widehat{\boldsymbol{A}(x - x^*)})_k\|_2^2\right]$$

$$\leq M\left(\|\boldsymbol{A}(x - x^*)\|_2^2 + \frac{1}{k^2}\|\boldsymbol{A}\|_F^2\|x - x^*\|_2^2\right)$$

Putting the value of $k = \sqrt{\kappa}$, and using strong convexity, $\|\boldsymbol{A}(x - x^*)\|_2^2 \geq \mu\|x - x^*\|_2^2$ we get that

$$\mathbb{E}\left[\|\nabla g(x) - \nabla g(x^*)\|_2^2\right] \leq M\left(\|\boldsymbol{A}(x - x^*)\|_2^2 + \|\boldsymbol{A}(x - x^*)\|_2^2\right) \leq 2M\|\boldsymbol{A}(x - x^*)\|_2^2$$

Note from Theorem 14, we know that the gradient update is of the following form.

$$x_{k+1} := x_k - \eta(\nabla g_i(x_k) - \nabla g_i(x_0)) + \eta\nabla f(x_0)$$

Note, the estimator $(\widehat{\boldsymbol{A}^\top \boldsymbol{A}x})_k$ uses $(\hat{a})_{c_i}$ and $(\widehat{a^\top x})_{c_i}$ estimators internally which both use $c_i = \sqrt{s_i}k$ coordinates. Hence, our estimator of $\nabla g_i(x_k) - \nabla g_i(x_0)$ can be implemented in $O(c_i)$ time when the $i$th row is chosen. Furthermore, the dense part $\nabla f(x_0) = \boldsymbol{A}^\top(\boldsymbol{A}x_0 - b)$ can be added in $O(1)$ time by separately maintaining the coefficient of this fixed vector in each $x_k$ and using it as necessary to calculate the $O(c_i)$ coordinates during each iteration. Consequently, we can bound the total expected time for implementing the iterations by

$$T = \sum_{i\in[n]} p_i c_i = \sum_{i\in[n]} \frac{\|a_i\|_2^2}{M}\left(1 + \frac{s_i}{c_i}\right)c_i = \sum_{i\in[n]} \frac{\|a_i\|_2^2}{M}(c_i + s_i) = \sum_{i\in[n]} \frac{\|a_i\|_2^2}{M}\left(\sqrt{\kappa} + \sqrt{s_i}\right)\sqrt{s_i}$$

Now, we know that $s_i \leq d$ and $\kappa \geq d$, hence, the first term in the above expression always dominates. Hence, we get the desired bound on $T$.

**Theorem 12 (Numerically Sparse Regression Runtime)** *For solving* $\epsilon$-*approximate regression (Definition 2), if* $\kappa \leq d^2$, *SVRG framework from Theorem 14 and the variance bound from Lemma 11 gives an algorithm with running time* $O\left(\left(\mathrm{nnz}(\boldsymbol{A}) + \sqrt{\kappa}\sum_{i\in[n]}\frac{\|a_i\|_2^2}{\mu}\sqrt{s_i}\right)\log\left(\frac{1}{\epsilon}\right)\right)$.

**Proof** The problem of solving regression $\min f(x)$ where $f(x) = \frac{1}{2}\|\boldsymbol{A}x - b\|_2^2$ can be solved by using the SVRG framework defined in Theorem 14 with the strong convexity parameter $\mu = \lambda_d(\boldsymbol{A}^\top \boldsymbol{A})$. Using the estimator for $\nabla f(x)$ defined in Lemma 11, we get the corresponding variance defined in Theorem 14 as $M$ i.e. $\sigma^2 = M$ where $M$ is as defined in Lemma 11. Therefore, according to Theorem 14, we can decrease the error by a constant factor in total number of iterations $O(\frac{M}{\mu})$. The expected time taken per iteration is $T = \frac{\sqrt{\kappa}}{M} \sum_{i \in [n]} \|a_i\|_2^2 \sqrt{s_i}$ as defined in Lemma 11. Now, since we know the expected number of iterations $O(\sigma^2/\mu)$ and expected time $O(T)$ per iteration, we can argue that with constant probability, the time taken is $O(T)$ and number of iterations is $O(\sigma^2/\mu)$ (upto constants) by using Markov inequality and hence, the total running time with constant probability will be $O(T\sigma^2/\mu)$ to decrease the error by constant multiplicative factor. Therefore, the total time taken would be $O\left(\mathrm{nnz}(\boldsymbol{A}) + \sqrt{\kappa} \sum_{i \in [n]} \frac{\|a_i\|_2^2}{\mu} \sqrt{s_i}\right)$ and thus, the total time taken to get an $\epsilon$-approximate solution to the problem would be $O\left(\left(\mathrm{nnz}(\boldsymbol{A}) + \sqrt{\kappa} \sum_{i \in [n]} \frac{\|a_i\|_2^2}{\mu} \sqrt{s_i}\right) \log\left(\frac{1}{\epsilon}\right)\right)$.

**Theorem 13 (Numerically Sparse Accelerated Regression Runtime)** *For solving $\epsilon$-approximate regression (Definition 2) if $\kappa \leq d^2$, SVRG framework from Theorem 14, acceleration framework from Theorem 15 and the variance bound from Lemma 11 gives an algorithm with running time*

$$O\left(\mathrm{nnz}(\boldsymbol{A})^{2/3} \kappa^{1/6} \left(\sum_{i \in [n]} \frac{\|a_i\|_2^2}{\mu} \sqrt{s_i}\right)^{1/3} \log(\kappa) \log\left(\frac{1}{\epsilon}\right)\right)$$

**Proof** Solving a regularized least squares problem i.e. $\min_x \|\boldsymbol{A}x - b\|_2^2 + \lambda\|x - x_0\|_2^2$ is equivalent to solving a problem with a modified matrix $\tilde{\boldsymbol{A}}$ with $n + d$ rows with the last $d$ rows having sparsity of 1 and rows $\tilde{a}_i = a_i$ if $i \leq n$ and 0 otherwise, $\tilde{s}_i = s_i$ if $i \leq n$ and 1 otherwise and $\tilde{\mu} = \mu + \lambda$ and therefore, by Theorem 12, the running time for solving the regularized regression upto constant accuracy will be

$$O\left(\mathrm{nnz}(\tilde{\boldsymbol{A}}) + \sqrt{\frac{\|\tilde{\boldsymbol{A}}\|_F^2}{\mu + \lambda}} \sum_{i \in [n+d]} \frac{\|\tilde{a}_i\|_2^2}{\mu + \lambda} \sqrt{\tilde{s}_i}\right)$$

which is equal to

$$O\left(\mathrm{nnz}(\boldsymbol{A}) + d + \sqrt{\frac{\|\boldsymbol{A}\|_F^2 + d\lambda^2}{\mu + \lambda}} \sum_{i \in [n]} \frac{\|a_i\|_2^2}{\mu + \lambda} \sqrt{s_i} + \sqrt{\frac{\|\boldsymbol{A}\|_F^2 + d\lambda^2}{\mu + \lambda}} \frac{d\lambda^2}{\mu + \lambda}\right).$$

Thus, the total running time for solving the unregularized problem will be, by Theorem 15

$$\tilde{O}\left(\left(\mathrm{nnz}(\boldsymbol{A}) + d + \sqrt{\frac{\|\boldsymbol{A}\|_F^2 + d\lambda^2}{\mu + \lambda}} \sum_{i \in [n]} \frac{\|a_i\|_2^2}{\mu + \lambda} \sqrt{s_i} + \sqrt{\frac{\|\boldsymbol{A}\|_F^2 + d\lambda^2}{\mu + \lambda}} \frac{d\lambda^2}{\mu + \lambda}\right) \sqrt{\frac{\lambda}{\mu}}\right)$$

where $\tilde{O}$ hides a factor of $\log\left(\frac{\lambda + 2\mu}{\mu}\right)$. Since $\lambda > 2\mu$ by the assumption of Theorem 15, we get $\lambda + \mu = O(\lambda)$, thus the total running time becomes

$$O\left(\left(\mathrm{nnz}(\boldsymbol{A}) + d + \sqrt{\frac{\|\boldsymbol{A}\|_F^2 + d\lambda^2}{\lambda}} \sum_{i \in [n]} \frac{\|a_i\|_2^2}{\lambda} \sqrt{s_i} + \sqrt{\frac{\|\boldsymbol{A}\|_F^2 + d\lambda^2}{\lambda}} \frac{d\lambda^2}{\lambda}\right) \sqrt{\frac{\lambda}{\mu}} \log\left(\frac{\lambda}{\mu}\right)\right)$$

Assuming $nnz(\boldsymbol{A}) \geq d$ and $\lambda^2 < \frac{\|\boldsymbol{A}\|_F^2}{d}$ since $\lambda$ should be less than the smoothness parameter of the problem. The running time becomes

$$O\left(\left(\mathrm{nnz}(\boldsymbol{A}) + \sqrt{\frac{\|\boldsymbol{A}\|_F^2}{\lambda}} \sum_{i \in [n]} \frac{\|a_i\|_2^2}{\lambda} \sqrt{s_i} + \sqrt{\frac{\|\boldsymbol{A}\|_F^2}{\lambda}} \frac{d\lambda^2}{\lambda}\right) \sqrt{\frac{\lambda}{\mu}} \log\left(\frac{\lambda}{\mu}\right)\right)$$

Since, $\lambda^2 \leq \frac{1}{d} \sum_{i \in [n]} \|a_i\|_2^2 \sqrt{s_i}$ by the assumption for this case as $s_i \geq 1$, hence, we get that the running time becomes

$$O\left(\left(\mathrm{nnz}(\boldsymbol{A}) + \sqrt{\frac{\|\boldsymbol{A}\|_F^2}{\lambda}} \sum_{i \in [n]} \frac{\|a_i\|_2^2}{\lambda} \sqrt{s_i}\right) \sqrt{\frac{\lambda}{\mu}} \log\left(\frac{\lambda}{\mu}\right)\right)$$

Balancing the two terms, we get the value of $\lambda = \left( \frac{\|A\|_F}{\mathrm{nnz}(A)} \sum_{i \in [n]} \|a_i\|_2^2 \sqrt{s_i} \right)^{2/3}$ which also satisfies our assumption on $\lambda$, hence the total running time becomes

$$O\left( \frac{\mathrm{nnz}(A)}{\sqrt{\mu}} \left( \frac{\|A\|_F}{\mathrm{nnz}(A)} \sum_{i \in [n]} \|a_i\|_2^2 \sqrt{s_i} \right)^{1/3} \log\left( \frac{\|A\|_F}{\mu^{3/2} \mathrm{nnz}(A)} \sum_{i \in [n]} \|a_i\|_2^2 \sqrt{s_i} \right) \right)$$

Thus, the running time for solving the system upto $\epsilon$ accuracy will be

$$O\left( \mathrm{nnz}(A)^{2/3} \kappa^{1/6} \left( \sum_{i \in [n]} \frac{\|a_i\|_2^2}{\mu} \sqrt{s_i} \right)^{1/3} \log(\kappa) \log\left( \frac{1}{\epsilon} \right) \right)$$

## C    Entrywise Sampling

In this section, we compute what bounds we get by first doing entrywise sampling on matrix $A$ to get $\tilde{A}$ and then running regression on $\tilde{A}$. Let us say we do entrywise sampling on the matrix $A \in \mathbb{R}^{n \times d}$ to obtain a matrix $\tilde{A} \in \mathbb{R}^{n \times d}$ with $s$ non-zero entries such that $\|A - \tilde{A}\|_2 \leq \epsilon$.
Then, we can write $\tilde{A} = A + E$ where $E \in \mathbb{R}^{n \times d}$ where $\|E\|_2 \leq \epsilon$
To get a bound on $\|A^\top A - \tilde{A}^\top \tilde{A}\|_2$

$$\|A^\top A - \tilde{A}^\top \tilde{A}\|_2 = \|A^\top A - (A + E)^\top (A + E)\|_2$$
$$= \| - A^\top E - E^\top A - E^\top E\|_2$$
$$\leq 2\sigma_{\max}(A)\epsilon + \epsilon^2$$

For $\epsilon \leq \frac{\lambda_{\min}(A^\top A)\epsilon'}{2\sigma_{\max}(A)}$, we get

$$\|A^\top A - \tilde{A}^\top \tilde{A}\|_2 \leq \lambda_{\min}(A^\top A)\epsilon'$$

and thus, we get that $\tilde{A}^\top \tilde{A}$ is a spectral approximation to $A^\top A$ i.e.

$$(1 - \epsilon')A^\top A \leq \tilde{A}^\top \tilde{A} \leq (1 + \epsilon')A^\top A$$

Thus, we can solve a linear system in $A^\top A$ to get $\delta$ multiplicative accuracy by solving $\log(\frac{1}{\delta})$ linear systems in $\tilde{A}^\top \tilde{A}$ upto constant accuracy and hence, the total running time will be $(\mathrm{nnz}(\tilde{A}) + \frac{\|\tilde{A}\|_F^2 s'}{\lambda_{\min}(\tilde{A}^\top \tilde{A})}) \log(\frac{1}{\delta})$ where $s'$ is the number of entries per row of $\tilde{A}$ i.e. we can find $x$ such that $\|x - x^*\|_{A^\top A} \leq \delta \|x_0 - x^*\|_{A^\top A}$ where $A^\top A x^* = c$.

Assuming uniform sparsity which is the best case for this appraoch and might not be true in general, we get the following running times by instantiating the above running time with different entry wise sampling results.

Using the results in [2] we get, $s' = O(\frac{\|A\|_F^2}{\epsilon^2})$ where $\epsilon = \frac{\lambda_{\min}(A^\top A)\epsilon'}{2\sigma_{\max}(A)}$ and hence $s' = \frac{\|A\|_F^2 \sigma_{\max}^2(A)}{\lambda_{\min}^2(A^\top A)\epsilon'^2}$ and hence a total running time of $\tilde{O}\left( \mathrm{nnz}(A) + \frac{\|A\|_F^4 \lambda_{\max}(A^\top A)}{\lambda_{\min}^3(A^\top A)} \right)$

Using the results in [1] we get, $s' = O(\frac{\sum_i \|A_{(i)}\|_1^2}{n\epsilon^2})$ and hence we get a total running time of $\tilde{O}\left( \mathrm{nnz}(A) + \frac{\sum_i \|A_{(i)}\|_1^2 \|A\|_F^2 \lambda_{\max}(A^\top A)}{n\lambda_{\min}^3(A^\top A)} \right)$ or $\tilde{O}\left( \mathrm{nnz}(A) + \frac{\sum_i s_i \|a_i\|_2^2 \|A\|_F^2 \lambda_{\max}(A^\top A)}{n\lambda_{\min}^3(A^\top A)} \right)$

Using the results in [5] we get, $s' = \sum_{ij} \frac{|A_{ij}|}{\sqrt{n}\epsilon}$ and hence, $s' = \frac{\sigma_{\max}(A)}{\lambda_{\min}(A^\top A)} \sum_{ij} \frac{|A_{ij}|}{\sqrt{n}\epsilon}$ and hence a total running time of $\tilde{O}\left( \mathrm{nnz}(A) + \frac{\sum_{ij} |A_{ij}| \|A\|_F^2 \sigma_{\max}(A)}{\sqrt{n}\lambda_{\min}^2(A^\top A)} \right)$ or $\tilde{O}\left( \mathrm{nnz}(A) + \frac{\sum_i \sqrt{s_i} \|a_i\|_2 \|A\|_F^2 \sqrt{\lambda_{\max}(A^\top A)}}{\sqrt{n}\lambda_{\min}^2(A^\top A)} \right)$

Table 3: Previous results in entry wise matrix sparsification. Given a matrix $\boldsymbol{A} \in \mathbb{R}^{n \times n}$, we want to have a sparse matrix $\tilde{\boldsymbol{A}} \in \mathbb{R}^{n \times n}$ satisfying $\|\boldsymbol{A} - \tilde{\boldsymbol{A}}\|_2 \leq \epsilon$. The first column indicates the number of entries in $\tilde{\boldsymbol{A}}$ (in expectation or exact). Note that this is not a precise treatment of entrywise sampling results since some results grouped together in the first row have different success probabilities and some results also depends on the ratio of the maximum and minimum entries in the matrix but this is the lower bound and we ignore details for simplicity since this suffices for our comparison.

| Previous entry wise sampling results | |
|---|---|
| Sparsity of $\tilde{\boldsymbol{A}}$ in $\tilde{O}$ | Citation |
| $n \frac{\|\boldsymbol{A}\|_F^2}{\epsilon^2}$ | [2, 11, 19, 18, 8, 14] |
| $\sqrt{n} \sum_{ij} \frac{|\boldsymbol{A}_{ij}|}{\epsilon}$ | [5] |
| $\sum_i \frac{\|a_i\|_1^2}{\epsilon^2} + \sqrt{\frac{\|\boldsymbol{A}\|_1^2}{\epsilon^2}}$ | [1] |