[Reviews · NeurIPS 2018]

Reviewer 1



The paper studies linear regression and top eigenvector computation. In particular, the focus of the work is improving the computational complexity by exploring the structure in the data. It is shown that if the data points are approximately sparse, the time complexity can be improved from O(d) to O(s), where "d" is the ambient dimension and "s" characterizes the sparsity. The major technique used is combining stochastic variance reduction gradient (SVRG) and coordinate sampling. ##Detailed Comments## - The paper is well-written and appears interesting. It turns out that the main contribution is an analysis of combining coordinate sampling and SVRG. Is this a new sampling technique, or has been studied elsewhere? - The paper considers very special problems, i.e. eigenvector computation and linear regression, which does offer a possibility of leveraging the structure of data. Can these results be extended to other problems, say a general optimization program? - From middle column of Table 1, the running time of this paper has a better dependence on the dimension, but at the same time depends on "sum ||a_i||^2" which can result in a worse complexity. The appearance of "lambda_1" also seems unusual. Same issue in Table 2. - Authors introduce too many notations that make the paper hard to follow. I have read the author response. I believe this is an interesting paper, but my concerns are not addressed in the author feedback. My first concern is the generality of the proposed method. It turns out that the technique applies only to quadratic loss functions, and in the feedback authors cannot point out applications beyond it. The second concern is the significance of theoretical results. The obtained computational complexity improves upon previous one in very limited regime (e.g. numerical sparsity is constant, hard sparsity is as large as dimension, condition number is d^2). It is not clear to me under which statistical setting can these assumptions be satisfied.

Reviewer 2



This paper proposes algorithms for faster run times for regression and top eigenvector computation of numerically sparse matrices. The authors show that improvements are achieved even when row norms and numerical sparsities are non-uniform and demonstrate that run times depend on input size and natural numerical measures of the matrix (e.g., eigenvalues and l-p norms). The problem of accelerating runtimes of regression and top eigenvector computation is relevant as data sets grow even larger and scaling up becomes mandatory for real-world application of machine learning. The problem can be reduced to a general finite sum optimization of minimizing a convex function f, decomposed into a sum of m functions (f_1, .., f_m). Past work has focused on `improving the dependence on the number of gradient evaluations of f_i that need to be performed (improving the dependence on m and other parameters). Here, the authors focus on what structural assumptions on f_i allow faster run times, i.e., by computing gradients of f_i approximately. They apply coordinate sampling techniques to stochastic variance reduced gradient descent (SVRG) to reach their results. Subsampling of the data matrix's row entries is done; however, this can change the shape of the variance, and the authors admit to this inherent issue. To mitigate this issue, they provide subsampling techniques to ensure a decrease in l2 error for small increases in samples taken per row. I have read the authors' feedback and thank them for their comments.

Reviewer 3



The paper proposes a method for solving least squares and top eigenvalue computation problems asymptotically faster than alternative approaches for numerically sparse matrices by proposing a new sampling method to approximate different quantities. My main problem with this paper is comparison to other approaches. I would love to see memory asymptotics, experimental comparison to see the constants involved, and comparison with Lanczos method (which is hard to analyse but which works well in practice). Also, line 239 says solving system with B is slow because of the condition number, and the condition number indeed get worse, but since one doesn’t have to solve the system exactly (see inexact inverse iteration) the time to solve it can still be ok. Plus, one can consider Jacobi-Davidson method which avoids this problem. Also, applying the method to A^t A leads to lost precision, it’s better to consider the augmented matrix [0 A A^T 0] UPD: I see that you focus on the theoretical side and propose a method with best worst case guarantees in this particular case. I personally would prefer to experimentally compare the method with the baselines anyway, but if others are OK without experiments, I would not object accepting the paper.

Reviewer 4



** Summary ** The paper studied two important problems in machine learning and numerical linear algebra: top eigenvector computation and l_2 regression. The authors introduce numerical sparsity’’ as a new parameter to measure the efficiency of algorithms for solving these problems. For a vector x, its numerical sparsity is defined to be |x|_1^2 / |x|_2^2. This quantity is a softer’’ version of sparsity and is never larger than the l_0 sparsity of x. The main new insight of this paper is that, it is possible to achieve better running time if we measure the efficiency of algorithms using this new parameter. To accomplish this goal, instead of devising completely new convex optimization methods, the authors develop new stochastic gradient oracle for these problems by importance sampling the matrix A (Section 3), whose variance bound depends on the numerical sparsity of rows of A, and then directly apply existing stochastic first order methods to solve l_2 regression and top eigenvector computation (Section 4). The main new technical result in this paper is an unbiased stochastic oracle to estimate A^T Ax for a matrix A and a vector x, which can serve as a stochastic gradient oracle when solving l_2 regression and top eigenvector computation. The variance of the oracle depends on the numerical sparsity of rows of A. I found this idea interesting and natural: consider a sparse vector a, if we slightly perturb a then its l_0 sparsity would be large. However, it is still possible to estimate such vector a accurately by (i) finding out heavy coordinates of a and (ii) importance sampling coordinates according to their magnitudes. The authors develop new stochastic oracle to estimate A^T Ax by formalizing these intuitions in Section 3. The main difficulty here is to relate the variance to the function error, and the authors achieve this goal by exploiting the numerical sparsity of rows of A. I found this sampling procedure interesting and may have applications in other machine learning / numerical linear algebra tasks. Overall, this is a good paper and I would recommend acceptance. ** Minor Comments ** Line 39: be achieved -> to be achieved Line 72: The definition of \lambda_min should be \lambda_d(A^T A) Line 79: (m/2) (a_i^T x - b_i) -> (m/2) (a_i^T x - b_i)^2 Line 212: In Algorithm 4, remove M in the denominator from the definition of M Line 378: how can we -> how we can Line 379: in smooth -> is smooth Line 380: remove the space after problems’’ Line 420: RHS of the equation should be (a^T x)^2 Line 429: The second inequality seems redundant Line 518: It seems you are assuming \kappa \ge d instead of \kappa \le d^2 in Lemma 11, since you want \sqrt{\kappa} \ge \sqrt{s_i}. This will also affect some related parts of the paper.